# Under the Shadow: Exploiting Opacity Variation for Fine-grained Shadow Detection

Xiaotian Qiao[1,2], Ke Xu[1], Xianglong Yang[2], Ruijie Dong[1], Xiaofang Xia[1*], Jiangtao Cui[1*]

[1]School of Computer Science and Technology, Xidian University, China
[2]Guangzhou Institute of Technology, Xidian University, China

## Abstract

Shadow characteristics are of great importance for scene understanding. Existing works mainly consider shadow regions as binary masks, often leading to imprecise detection results and suboptimal performance for scene understanding. We demonstrate that such an assumption oversimplifies light-object interactions in the scene, as the scene details under either hard or soft shadows remain visible to a certain degree. Based on this insight, we aim to reformulate the shadow detection paradigm from the opacity perspective, and introduce a new fine-grained shadow detection method. In particular, given an input image, we first propose a shadow opacity augmentation module to generate realistic images with varied shadow opacities. We then introduce a shadow feature separation module to learn the shadow position and opacity representations separately, followed by an opacity mask prediction module that fuses these representations and predicts fine-grained shadow detection results. In addition, we construct a new dataset with opacity-annotated shadow masks across varied scenarios. Extensive experiments demonstrate that our method outperforms the baselines qualitatively and quantitatively, enhancing a wide range of applications, including shadow removal, shadow editing, and 3D reconstruction.

## 1 Introduction

Shadow is a natural phenomenon when objects obstruct light sources either fully or partially, resulting in intensity and color variations across certain regions. These shadow characteristics play vital roles for scene understanding. The presence of shadows can degrade image quality and impede numerous vision and graphics tasks, including image editing [48], object detection [33], and visual tracking. Furthermore, shadow analysis allows us to interpret semantics and geometry of a scene [31], *e.g.*, object shapes, light source positions, and object relationships. However, accurately detecting shadow characteristics (*e.g.*, boundary, opacity, etc.) in the scene can be very challenging. Shadows exhibit diverse characteristics due to scene factors like lighting conditions, surface properties, and occluder configurations, resulting in various shadow types, from sharp to diffuse.

While advanced shadow detection methods have been proposed, they often regard shadows as binary masks or assume uniform opacity within shadow regions [2, 17, 49, 42]. We argue that such a binary or simplified lighting paradigm oversimplifies light-object interactions, as the scene details under either hard or soft shadows retain varying degrees of visibility. As shown in Figure 1, existing methods [54, 55] fail to capture the nuanced opacity variations and suffer from position and shape estimation artifacts, undermining downstream applications like shadow removal and editing that rely on precise shadow characteristics. A fine-grained formulation of the shadow regions remains underexplored, which is vital for scene understanding.

---

*Joint corresponding authors.

39th Conference on Neural Information Processing Systems (NeurIPS 2025).

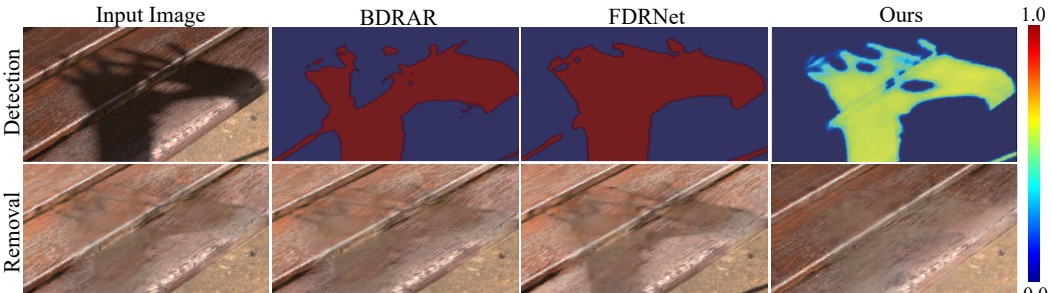

Figure 1: Fine-grained shadow detection. Given an input image (1st row), existing methods [17, 54, 55] predict binary shadow masks (1st, 2nd and 3rd columns), yielding inaccurate shadow detection results. In contrast, we propose to detect fine-grained shadow characteristics (4th column) that capture subtle opacity variations, with benefits for the downstream applications like shadow removal [12] (2nd row).

We observe that scene details under both umbra (fully occluded) and penumbra (partially occluded) regions retain varying degrees of visibility. The variation of shadow opacity caused by factors like occluder distance or light source intensity, could offer useful scene contextual cues. Take the input image in Figure 1 as an example. The subtle gradation of the shadow boundary suggests occluder distance, while umbra darkness and transparency reflect illumination intensity.

Inspired by the above observation, we propose a learning-based framework to predict a continuous map that indicates the position, boundary, and opacity variation of shadow regions. Our framework contains three main modules. First, given the input shadow image, we propose a Shadow Opacity Augmentation (SOA) module to generate multiple images with varied shadow opacities. Each augmented image and the original image are grouped as an image pair. Second, for each image pair, we propose a Shadow Feature Separation (SFS) module to disentangle position and opacity representations of the shadows separately. Third, we propose an Opacity Mask Prediction (OMP) module that fuses the learned representations and predicts a continuous shadow map as the output.

To support detailed analysis of the proposed paradigm, we further construct a new Fine-grained Shadow Detection (*i.e.*, FSD) dataset. In particular, it contains 2,653 images with opacity-annotated shadow masks across diverse scenarios, including different light source intensities and numbers, indoor and outdoor scenes. We conduct extensive qualitative and quantitative experiments on public datasets and the proposed dataset. Results show that our approach outperforms baselines in capturing fine-grained shadow characteristics, enhancing downstream scene understanding applications.

To sum up, we make the first attempt to investigate fine-grained shadow detection by exploiting opacity variations that are ignored by existing shadow detection and removal methods. We propose a new shadow detection method by explicitly capturing shadow position and opacity characteristics, and construct a new dataset with shadow opacity annotations across varied scenarios. Experimental results show that our method offers a promising shadow detection pipeline that can be applied across various applications, including shadow removal, shadow editing, and 3D reconstruction.

## 2   Related Work

**Shadow Detection.**   Early research works rely on physical assumptions or heuristic cues. For instance, Lin et al. [29] adopted physical models assuming illumination invariance, while others analyzed chromatic variations in shadow regions using chromaticity [8, 9, 25], gradient [8, 25, 53], and intensity cues [39, 13, 53]. Huang et al. [20] trained a shadow detector by feeding edge features into a support vector machine. Guo et al. [14] computed illumination features of segmented regions and constructed a graph-based classifier by leveraging inter-region relationships. All these methods depend on handcrafted features, making it difficult to detect shadows accurately in complex scenes.

Deep learning has significantly improved the shadow detection performance [24]. Qu et al. [35] combined fully connected networks with patch-CNN refinement. Nguyen et al. [34] stabilized training using adversarial networks, and Hu et al. [17, 19, 54] leveraged spatial contexts. Wang et

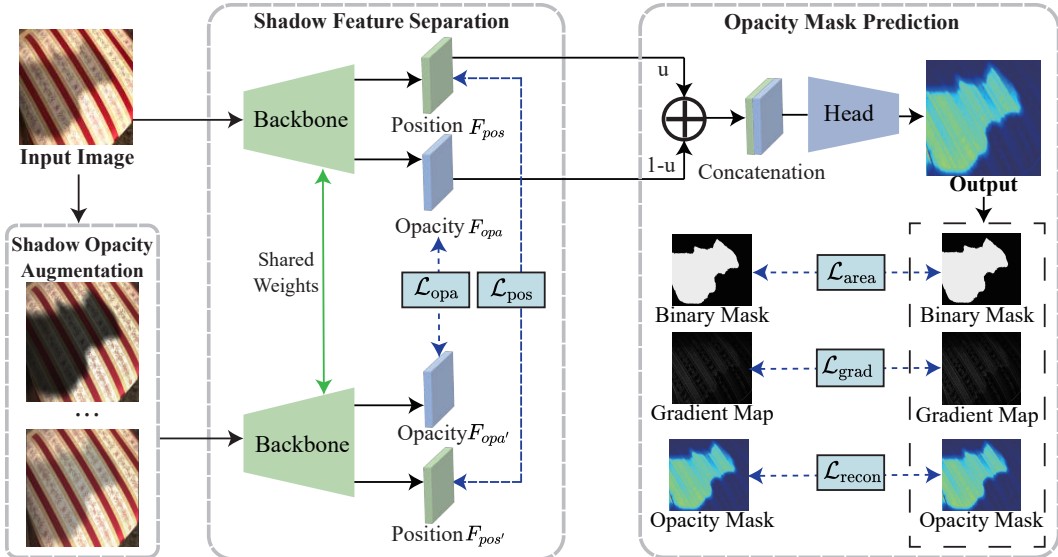

Figure 2: The overall pipeline of our fine-grained shadow detection.

al. [43, 6] and Zheng et al. [51] improved accuracy via adversarial networks and distraction modeling, respectively. Chen et al. [1] incorporated edge-assisted detection. Recent works [49, 36] further addressed annotation subjectivity and enhanced features via multi-color-space training.

However, these methods tend to produce binary masks, ignoring fine-grained shadow opacity. While Wang et al. [45] also noted the limitations of the binary shadow mask, they only generate soft masks implicitly as intermediate results for shadow removal. In contrast, our approach models the shadow opacity variations explicitly, benefiting various downstream scene understanding applications.

**Shadow Matte Estimation.** Shadow matte estimation aims to model the appearance of shadows in natural scenes. Chuang et al. [2] proposed matte shadows and defined a shadow compositing formula. Wu et al. [46] optimized the shadow compositing formulation and proposed a method to eliminate shadows using the shadow matte. Lu et al. [32] introduced a generic framework capable of matting objects with associated shadows. Lin et al. [28] further improved this to effectively matte objects and their corresponding shadows in 3D scenes.

However, shadow mattes involve additional colors and textures, making it difficult to be utilized directly for downstream applications. In contrast, our opacity-aware shadow mask formulation allows for more flexible and controllable shadow removal and editing results.

## 3   Approach

Given an input shadow image $I_s$, our goal is to predict a continuous opacity mask $M_\alpha$ representing the shadow region. The shadow formulation is defined as follows:

$$I_s = I_f \cdot (1 - M_\alpha) + M_\alpha \cdot S, \tag{1}$$

where $I_f$ is the shadow-free image. $S$ represents the shadow color, which could be a constant value. $M_\alpha$ represents the continuous shadow opacity, ranging from 0 (fully transparent) to 1 (fully opaque).

Figure 2 shows the overall pipeline of our method, which contains a shadow opacity augmentation (SOA) module, a shadow feature separation (SFS) module, and an opacity mask prediction (OMP) module. We first apply opacity augmentation by adjusting the opacity of shadow regions in the input image. We then pass original and augmented images through the weights-shared backbone to obtain position and opacity features. The position features $F_{pos}$ and $F'_{pos}$ should be the same, while the opacity features $F_{opa}$ and $F'_{opa}$ should reflect the augmentation difference. Finally, both position and opacity features are fused through a shadow detection head to output the continuous opacity map.

### 3.1 Shadow Opacity Augmentation (SOA) Module

A trivial solution is to use an auto-encoder architecture to predict the opacity shadow mask directly. However, it performs poorly in handling diverse and complex shadow scenarios. To fully leverage the shadow opacity characteristics, we perform shadow augmentation on the input images by randomly augmenting the opacity of shadow regions.

Specifically, given an input image $I_s$ and its opacity shadow mask $M_\alpha$, we modify the opacity of the shadow regions by randomly sampling the parameter $\beta$ to generate an augmented shadow image $\hat{I}_s$. Note that the SOA module is only used for training, where $I_s$ and $\hat{I}_s$ are grouped as an image pair.

### 3.2 Shadow Feature Separation (SFS) Module

Given each image pair that indicates the same scene with different shadow opacities as input, we extract the position and opacity features separately. For the original image $I_s$, we introduce a backbone to extract the position feature $F_{pos}$ and the opacity feature $F_{opa}$. Similarly, for the augmented shadow image $\hat{I}_s$, we use the weight-shared backbone to extract $\hat{F_{pos}}$ and $\hat{F_{opa}}$ correspondingly.

Since the shadow positions in the image pair remain unchanged, the position features $F_{pos}$ and $\hat{F_{pos}}$ should be identical theoretically. Meanwhile, the opacity features $F_{opa}$ and $\hat{F_{opa}}$ should exhibit the differences caused by the opacity augmentation parameter $\beta$.

### 3.3 Opacity Mask Prediction (OMP) Module

Given the set of the position feature $F_{pos}$ and the opacity feature $F_{opa}$, we fuse these representations and predict the continuous shadow opacity map as the fine-grained shadow detection result. In particular, we first fuse the position feature $F_{pos}$ and the opacity feature $F_{opa}$ as follows:

$$F_s = \mu F_{pos} + (1 - \mu) F_{opa}. \tag{2}$$

We then employ a cumulative learning strategy [52] to optimize the fused shadow feature $F_s$ through a weight selection strategy. For training epoch $T$ and the total training epochs $T_{max}$, $\mu$ is defined as:

$$\mu = 1 - \left(\frac{T}{T_{\max}}\right)^\eta, \tag{3}$$

where $\eta$ is a hyperparameter controlling the decay rate of $\mu$ during the training phase. Finally, the optimized shadow features are passed through a detection head to predict the opacity mask.

### 3.4 Training

For the SFS module, we use $\mathcal{L}_{pos}$ to constrain the consistency of the position features:

$$\mathcal{L}_{\text{pos}} = \text{MAE}(F_{pos}, \hat{F_{pos}}), \tag{4}$$

where MAE is a mean-absolute-error loss function. The shadow opacity features $F_{opa}$ and $\hat{F_{opa}}$ should be capable of predicting the transparency augmentation parameter $\beta$. To achieve this, we subtract $F_{opa}$ from $\hat{F_{opa}}$ to predict $\beta$:

$$\mathcal{L}_{\text{opa}} = |\text{MLP}(F_{opa} - \hat{F_{opa}}) - \beta|, \tag{5}$$

where MLP contains a global average pooling layer and fully connected layers.

For the OMP module, to mitigate the computational difficulty arising from the varying proportions of shadow and non-shadow regions in each image, we employ $\mathcal{L}_{area}$ to calculate the difference for shadow and non-shadow regions and multiply by their weights:

$$\mathcal{L}_{\text{area}} = \frac{\sum_i \text{L}_s\left(A_i^{\text{pred}}, A_i^{\text{gt}}\right) \cdot [A_i^{\text{gt}} = 0]}{\sum_i [A_i^{\text{gt}} = 0]} + \frac{\sum_i \text{L}_s\left(A_i^{\text{pred}}, A_i^{\text{gt}}\right) \cdot [A_i^{\text{gt}} \neq 0]}{\sum_i [A_i^{\text{gt}} \neq 0]}, \tag{6}$$

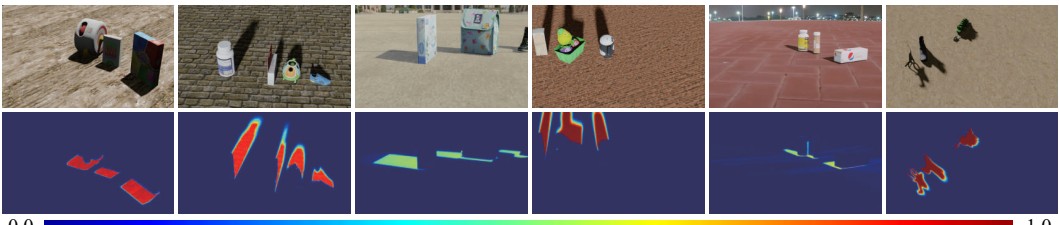

0.0                   1.0

Figure 3: Example images with opacity mask annotations in our FSD dataset.

where $A_i^{\text{pred}}$ represents the predicted value of the i-th pixel, and $A_i^{\text{gt}}$ represents the true value of the i-th pixel. The definition of $L_s$ is:

$$L_s(\text{pred},\text{gt}) = \begin{cases} 0.5 \cdot (\text{pred} - \text{gt})^2 & \text{if } |\text{pred} - \text{gt}| < 1 \\ |\text{pred} - \text{gt}| - 0.5 & \text{otherwise.} \end{cases} \tag{7}$$

To achieve smooth prediction results that are close to the real shadow values, we also use gradient loss $\mathcal{L}_{grad}$ as follows:

$$\mathcal{L}_{\text{grad}} = (\|\nabla_x\text{pred} - \nabla_x\text{gt}\| + \|\nabla_y\text{pred} - \nabla_y\text{gt}\|), \tag{8}$$

where $\nabla_x$ and $\nabla_y$ represent the gradients in the x and y directions, respectively.

We further utilize $\mathcal{L}_{recon}$ to address the overall discrepancy as follows:

$$\mathcal{L}_{\text{recon}} = \frac{1}{N}\sum_{i=1}^{N}(pred - gt)^2. \tag{9}$$

In summary, we train our model with the following losses:

$$\mathcal{L}_{\text{all}} = \lambda_{\text{pos}}\mathcal{L}_{\text{pos}} + \lambda_{\text{opa}}\mathcal{L}_{\text{opa}} + \lambda_{\text{area}}\mathcal{L}_{\text{area}} + \lambda_{\text{grad}}\mathcal{L}_{\text{grad}} + \lambda_{\text{recon}}\mathcal{L}_{\text{recon}}, \tag{10}$$

where $\lambda_{pos}$ $\lambda_{opa}$ $\lambda_{area}$ $\lambda_{grad}$, and $\lambda_{\text{recon}}$ are the weighting parameters.

## 4 Dataset

A number of shadow datasets [35, 43, 18] have been proposed in the community, advancing the progress of shadow detection and removal tasks. However, as shown in Table 1, these datasets suffer from shortcomings like lacking scenes with weak lighting, non-point light sources, or multi-source soft shadows. Most importantly, there are no publicly available shadow datasets considering opacity characteristics.

| Dataset | Opacity Mask | Binary Mask | Object Association | Shadow Free |
|---|---|---|---|---|
| SRD [35] | ✗ | ✗ | ✗ | ✓ |
| SBU [41] | ✗ | ✓ | ✗ | ✗ |
| ISTD [43] | ✗ | ✓ | ✗ | ✓ |
| USR [30] | ✗ | ✗ | ✗ | ✓ |
| SOBA [44] | ✗ | ✓ | ✓ | ✗ |
| DESOBA [16] | ✗ | ✓ | ✓ | ✓ |
| RdSOBA [38] | ✗ | ✓ | ✓ | ✓ |
| FSD (ours) | ✓ | ✓ | ✓ | ✓ |

Table 1: A taxonomy of shadow datasets.

To increase the diversity of the images, we use Blender to construct a new synthetic fine-grained shadow detection (i.e., FSD) dataset with the following guidelines:

- **Object**. We select common and diverse object categories from daily life (e.g., person, animals, shoes), and place 1 to 5 objects with randomized positions in each scene.
- **Scene**. The shadow images were captured from different camera angles across various indoor and outdoor scenarios, with all cameras configured at a 50mm focal length, automatic sensor settings, and randomized positions.
- **Light Source**. To enhance diversity, the number, position, and intensity of light sources are considered, with their positions and brightness levels being randomly sampled.

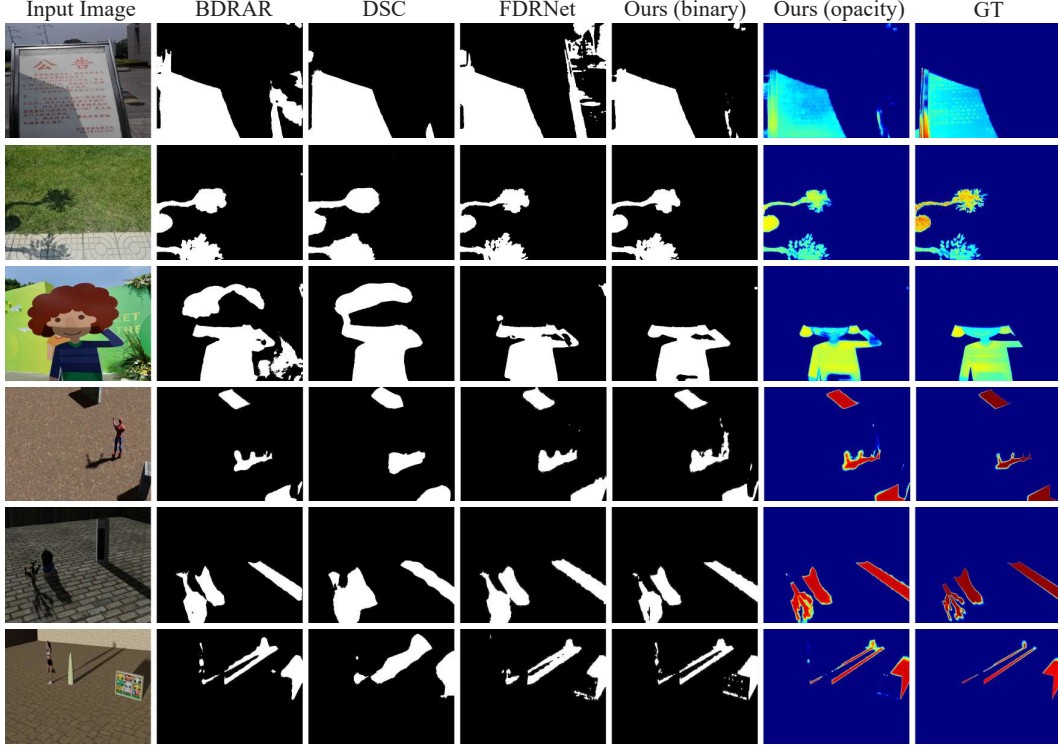

Figure 4: Qualitative comparisons of shadow detection on the ISTD [43] and FSD datasets. Given the input image (1st column), we show the baseline results (2nd to 4th columns), our binary mask(5th column), our opacity mask (6th column), and ground truth (7th column), respectively.

In summary, FSD comprises 2,653 scenes with different object, scene, and light source properties. Each scene contains varied camera positions, light intensities, object numbers, and categories, resulting in 7,701 object-shadow pairs. Some examples with varying shadow opacities are shown in Figure 3. Please refer to the supplementary material for additional details.

## 5 Experiments

### 5.1 Experimental Setup

**Implementation Details.** All experiments are conducted on a single NVIDIA RTX 4090 GPU. We use EfficientNet-B3 [37] as the backbone network and ImageNet [5] for weights initialization. Our model is trained using the Adam optimizer for 20 epochs, with an initial learning rate of 5e-4, dynamically adjusted through an exponential decay strategy (decay rate of 0.7). During training, the input images are resized to $400 \times 400$ with a batch size of 12. The random horizontal flipping operation is further used for data augmentation. The weight parameters $\lambda_{pos}$, $\lambda_{opa}$, $\lambda_{area}$, $\lambda_{grad}$, and $\lambda_{\text{recon}}$ are set to 1, 1, 10, 10, and 10, respectively.

**Datasets.** We conduct experiments on both the proposed FSD dataset and the existing ISTD [43] dataset, which contains 1,330 training images and 540 test images. Furthermore, we utilize two shadow removal datasets, *i.e.*, ISTD+ [26] and SRD [35], for downstream applications. The ground truth shadow masks in SRD are obtained by utilizing a connectivity-based optimization strategy.

**Compared Methods.** As this is the first work for fine-grained shadow detection that considers opacity, to make fair comparisons, we convert our output to binary shadow masks, and compare our method with existing shadow detection methods, including SDCM [56], FDRNet [55], MTMT [1], DSD [51], DSC [17], BDRAR [54], ST-CGAN [43], scGAN [34], stacked-CNN [41].

| Dataset | Method | BER ↓ | Shadow ↓ | Non Shad.↓ |
|---------|--------|-------|----------|------------|
| ISTD | stacked-CNN [41] | 8.60 | 7.69 | 9.23 |
|  | scGAN [34] | 4.70 | 3.22 | 6.18 |
|  | ST-CGAN [43] | 3.85 | 2.14 | 5.55 |
|  | BDRAR [54] | 2.69 | **0.50** | 4.87 |
|  | DSC [17] | 3.42 | 3.85 | 3.00 |
|  | DSD [51] | 2.17 | 1.36 | 2.98 |
|  | MTMT [1] | 1.72 | 1.36 | 2.08 |
|  | FDRNet [55] | 1.55 | 1.22 | 1.88 |
|  | SDCM [56] | 1.40 | 1.02 | 1.78 |
|  | Ours | **1.32** | 0.96 | **1.67** |
| FSD | DSC [17] | 4.46 | 4.99 | 3.93 |
|  | BDRAR [54] | 4.43 | 7.74 | **1.12** |
|  | ST-CGAN [43] | 4.19 | 4.63 | 3.76 |
|  | FDRNet [55] | 3.01 | 2.58 | 3.43 |
|  | Ours | **1.79** | **1.94** | 1.64 |

Table 2: Quantitative comparison of the proposed method with baselines on the ISTD [43] and FSD datasets. The best and second-best performances are marked in bold and underlined.

**Evaluation Metrics.** We evaluate the quality of the predicted fine-grained shadow mask from different aspects. We use the Balanced Error Rate (BER) metric to measure the position and boundary of the shadows. In addition, to evaluate the accuracy of predicted shadow opacity, we propose a new metric termed Weighted Shadow Error (WSE). Since non-shadow areas typically dominate most images, commonly used metrics like RMSE may overlook subtle opacity variations within shadow regions. As a result, WSE incorporates a weighted strategy that emphasizes shadow region accuracy by computing a region-aware RMSE, thereby offering a more balanced assessment of fine-grained shadow detection performance.

$$\text{WSE} = \left( \frac{1 - \frac{\sum_{i=1}^{N}[A_i^{\text{gt}} \neq 0]}{N}}{\frac{\sum_{i=1}^{N}[A_i^{\text{gt}} \neq 0]}{N}} \right) \cdot \text{RMSE}_{\text{sdr}} + \text{RMSE}_{\text{nsdr}}, \tag{11}$$

where $A_i^{\text{gt}}$ represents the alpha value of shadow for each pixel. $\text{RMSE}_{\text{sdr}}$ and $\text{RMSE}_{\text{nsdr}}$ are the root mean square error of the shadow and non-shadow regions, respectively.

## 5.2 Results

**Qualitative Evaluation.** Visual comparisons on the ISTD [43] and FSD datasets are presented in Figure 4. Different from existing methods that predict only binary masks for shadow regions, our approach can further capture the degree of shadow degradation in the scene accurately. Moreover, by explicitly modeling shadow opacity, our method substantially reduces both false positives and false negatives. Take the second row as an example. Our method delivers fine-grained detection results around the shadow boundaries of the tree leaves, demonstrating clear advantages in predicting precise shadow characteristics.

**Quantitative Evaluation.** We compare the performance of our method with state-of-the-art shadow detection methods in Table 2. As the existing shadow datasets do not contain ground truth opacity annotations, we extract the brightness difference between shadowed and shadow-free images as the ground truth value for the opacity shadow mask. Our method achieves the best or second-best BER scores among all compared methods, significantly validating the effectiveness of the shadow opacity guidance. Note that our method achieves a greater performance gain on the FSD dataset than on the ISTD dataset. The primary reason is that our FSD dataset includes a diverse range of challenging cases featuring soft shadow boundaries and varied opacity levels. Thus, these results highlight the strength of our method in handling diverse shadow characteristics in complex scenarios.

| Method | BER↓ | RMSE↓ | WSE↓ |
|---|---|---|---|
| AutoEncoder | 4.01 | 0.0919 | 10.49 |
| w/o $F_{opa}$ | 1.97 | - | - |
| w/o $F_{pos}$ | 2.82 | **0.0637** | 5.93 |
| w/o SOA | 2.16 | 0.0744 | 6.02 |
| Ours (full) | **1.78** | 0.0699 | **5.78** |

Table 3: Ablation study of different components.

| Method | BER↓ | RMSE↓ | WSE↓ |
|---|---|---|---|
| w/o $\mathcal{L}_{grad}$ and $\mathcal{L}_{opa}$ | 3.27 | - | - |
| w/o $\mathcal{L}_{grad}$ and $\mathcal{L}_{area}$ | 2.34 | 0.0725 | 6.81 |
| w/o $\mathcal{L}_{grad}$ | 2.03 | 0.0691 | 6.74 |
| w/o $\mathcal{L}_{area}$ | 1.86 | 0.0682 | 6.03 |
| Ours (full) | **1.78** | **0.0669** | **5.78** |

Table 4: Ablation study of different losses.

## 5.3 Ablation Study

We evaluate the impact of different components, losses, and training strategies of our pipeline on the FSD dataset. Further analysis on different datasets are provided in the supplementary material.

**Component Analysis.** We first introduce a naive baseline (i.e., AutoEncoder) with a simple encoder and decoder architecture. We then conduct two ablations for the SFS module. Specifically, we remove the position feature branch (i.e., w/o $F_{pos}$) and the opacity feature branch (i.e., w/o $F_{opa}$) individually. We further remove the SOA module (i.e., w/o SOA) to examine the effect of the shadow opacity augmentation strategy. As shown in Table 3, these components play essential roles in resolving ambiguities in fine-grained shadow characteristics.

| Method | BER↓ | RMSE↓ | WSE↓ |
|---|---|---|---|
| w/o cum. | 2.18 | 0.1505 | 10.83 |
| $\eta = 0.3$ | 1.93 | **0.0635** | 6.21 |
| $\eta = 0.4$ | 1.83 | 0.0984 | 6.16 |
| $\eta = 0.6$ | 1.80 | 0.0675 | 6.17 |
| $\eta = 0.5$ | **1.78** | 0.0669 | **5.78** |

Table 5: Ablation study of the training strategy.

**Loss Analysis.** The effects of different losses are examined in Table 4. We first remove $\mathcal{L}_{grad}$ and $\mathcal{L}_{opa}$ to study the importance of shadow opacity guidance. We then remove $\mathcal{L}_{grad}$ and $\mathcal{L}_{area}$ separately or totally. The results show that compared to using $\mathcal{L}_{recon}$ alone, the integration of $\mathcal{L}_{grad}$, $\mathcal{L}_{area}$, and $\mathcal{L}_{opa}$ enhance the fine-grained shadow detection performance significantly.

**Training Analysis.** We also ablate the cumulative learning strategy in Table 5. We retrain the model without using the cumulative learning strategy (i.e., w/o cum.), and use different $\eta$ values to control the decay rate of $\mu$. The results show that $\eta = 0.5$ achieves the best performance across all metrics.

| | Methods | Input Masks | All | | | Shadow | | | Non-Shadow | | |
|---|---|---|---|---|---|---|---|---|---|---|---|
| | | | PSNR↑ | SSIM↑ | MAE↓ | PSNR↑ | SSIM↑ | MAE↓ | PSNR↑ | SSIM↑ | MAE↓ |
| ISTD+ | DC-ShadowNet [22] | NA | 25.03 | 0.926 | 7.77 | 31.06 | 0.976 | 12.62 | 27.03 | 0.961 | 6.82 |
| | DeS3 [23] | NA | 31.38 | 0.958 | 3.94 | 36.49 | 0.989 | 6.56 | 34.70 | 0.972 | 3.40 |
| | BMNet [57] | FDRNet [55] | 32.41 | 0.962 | 3.40 | 36.59 | 0.989 | 6.65 | 35.96 | 0.978 | 2.83 |
| | HomoFormer [47] | FDRNet [55] | 32.41 | 0.953 | 3.51 | 38.84 | 0.991 | 5.31 | 34.58 | 0.966 | 3.17 |
| | ShadowDiffusion [12] | FDRNet [55] | **34.08** | 0.968 | 3.12 | **40.12** | 0.992 | 5.15 | 36.66 | 0.978 | 2.74 |
| | ShadowDiffusion [12] | Ours | 34.06 | **0.969** | **3.11** | 39.49 | 0.992 | 5.28 | **37.22** | **0.981** | **2.68** |
| SRD | DC-ShadowNet [22] | NA | 31.53 | 0.955 | 4.65 | 34.00 | 0.975 | 7.70 | 35.53 | 0.981 | 3.65 |
| | DeS3 [23] | NA | 34.11 | 0.968 | 3.56 | 37.91 | 0.986 | 5.27 | 37.45 | 0.984 | 3.03 |
| | SAM-helps-shadow [50] | NA | 30.72 | 0.952 | 4.79 | 33.94 | 0.979 | 7.44 | 33.85 | 0.981 | 3.74 |
| | ShadowFormer [11] | DHAN [3] | 32.46 | 0.957 | 4.28 | 35.55 | 0.982 | 6.14 | 36.82 | 0.983 | 3.54 |
| | DHAN [3] | DHAN [3] | 30.51 | 0.949 | 5.67 | 33.67 | 0.978 | 8.94 | 34.79 | 0.979 | 4.80 |
| | BMNet [57] | DHAN [3] | 31.69 | 0.956 | 4.46 | 35.05 | 0.981 | 6.61 | 36.02 | 0.982 | 3.61 |
| | Inpaint4Shadow [27] | DHAN [3] | 33.27 | 0.967 | 3.81 | 36.73 | 0.985 | 5.70 | 36.70 | 0.985 | 3.27 |
| | Homoformer [47] | DHAN [3] | **35.37** | 0.972 | 3.33 | 38.81 | 0.987 | 4.25 | **39.45** | **0.988** | **2.85** |
| | ShadowDiffusion [12] | DHAN [3] | 34.73 | 0.970 | 3.63 | 38.72 | 0.987 | 4.98 | 37.78 | 0.985 | 3.44 |
| | ShadowDiffusion [12] | FDRNet [55] | 34.23 | 0.972 | 3.50 | 37.31 | 0.985 | 5.04 | 38.61 | 0.986 | 2.90 |
| | ShadowDiffusion [12] | Ours | 34.84 | **0.974** | **3.27** | **42.06** | **0.995** | 3.32 | 36.45 | 0.986 | 3.28 |

Table 6: Quantitative comparisons with the shadow removal methods on ISTD+ [26] and SRD [35] datasets. NA indicates that no mask input is required. The best and the second results are highlighted in bold and underlined, respectively.

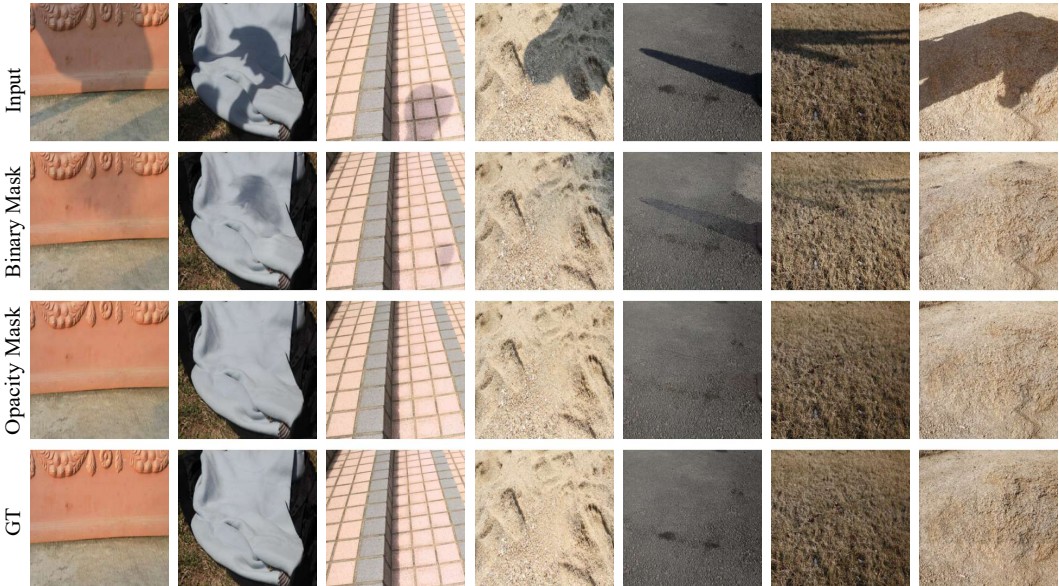

Figure 5: Shadow removal results. Given the input image (1st row), we show shadow removal results from ShadowDiffusion [12] with binary masks (2nd row), our opacity masks (3rd row), and the ground truth (4th row).

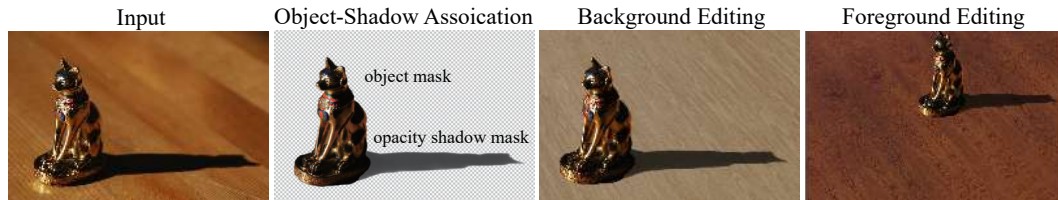

Input      Object-Shadow Assoication      Background Editing      Foreground Editing

object mask

opacity shadow mask

Figure 6: Shadow editing results. Given the input image (1st column), we extract the opacity shadow mask with the corresponding object (2nd column). Multiple editing operations can be further conducted, and corresponding fine-grained shadow characteristics will be changed adaptively. For example, replacing the background (3rd column) or changing the foreground object (4th column).

## 5.4 Applications

Benefiting from our model for fine-grained shadow detection, we explore three downstream applications here. Please refer to the supplementary material for additional details and results.

**Shadow Removal.** The opacity shadow mask provides rich information by explicitly encoding pixel-wise attenuation coefficients, enabling precise characterization of shadow degradation levels on the background. To validate its practical utility, we reformulate the training pipeline of ShadowDiffusion [12] by replacing its binary shadow mask input with our continuous opacity map. Such an adaptation can dynamically adjust denoising strength based on shadow opacity and preserve natural illumination transitions at shadow boundaries through opacity-guided attention modulation.

Quantitative comparisons against existing shadow removal methods are presented in Table 6. The results show that the shadow removal model trained with the opacity shadow mask achieves substantial improvements. Note that our method obtains greater performance gains on shadow regions than on non-shadow regions. The primary reason is that our method is specifically designed to handle shadow opacity, while non-shadow regions dominate most images, masking shadow-specific improvements. For example, on the SRD dataset [35] that contains more complex scenes and soft shadows, the PSNR value for shadow regions shows substantial enhancement, validating the effectiveness of the opacity guidance strategy. Qualitative comparisons in Figure 5 further highlight our advantages in shadow removal visual quality, particularly for shadows with light-color or blurred boundaries.

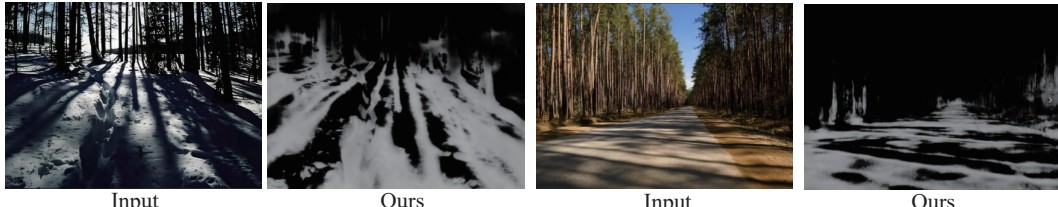

| Input | Ours | Input | Ours |

Figure 7: Failure cases. Our model may fail on images with complex shadow interactions.

**Shadow Editing.** Shadows often complicate image editing by hindering consistent manipulation of objects and their associated shading. However, by employing the proposed opacity shadow mask, we can simplify such a process by enabling synchronized adjustments of shadows and objects. As shown in Figure 6, the opacity shadow mask can be treated as a detachable shadow layer and edited alongside the object. Specifically, given an input image, we utilize the proposed model to extract an opacity shadow mask with the corresponding object. Multiple editing operations can be further conducted, and corresponding fine-grained shadow characteristics will be changed adaptively. For example, we can replace the background in the input image with a new one, or modify the position and size of the corresponding object in the image.

**3D reconstruction.** Fine-grained shadow opacity variations (e.g., penumbra width and transparency gradients) provide critical cues about scene geometry, such as occluder distance and light source intensity, which are highly valuable for 3D reconstruction. Following the shadow-driven neural rendering framework [40], we apply our continuous opacity maps, rather than binary masks from FDRNet [55] used before, for the 3D reconstruction application. Our method achieved RMSE values of 0.00957 and 0.00531 for the Bunny and CUBE scenes, respectively, significantly lower than FDRNet [55]'s 0.01074 and 0.00777. These results demonstrate that leveraging fine-grained shadow properties can enhance geometric reconstruction accuracy, particularly under texture-less or complex lighting conditions.

**Discussion.** The transparency information embedded in the opacity shadow mask provides additional benefits for scene understanding. First, it enables precise modeling of shadow-object interactions by quantifying the opacity gradient at shadow boundaries, which can guide existing shadow generation models to synthesize more realistic soft shadows with adaptive opacity. The pixel-wise opacity map serves as a physical prior for illumination decomposition, allowing joint optimization of shadow removal and scene relighting tasks. In dynamic scenes with moving objects, the temporal consistency of opacity values can be leveraged to enhance video shadow stabilization algorithms, reducing flickering artifacts during editing.

Although our model works well in various shadow scenarios, it may fail in some challenging cases. As illustrated in Figure 7, it would be difficult for our model to predict accurate shadow opacity when multiple shadow interactions exist in complex scenarios. A potential solution to this problem is to adopt the instance shadow strategy to distinguish specific object-shadow associations. As future work, we would like to explore more shadow characteristics to boost the scene understanding ability.

# 6   Conclusion

In this paper, we make the first attempt to investigate fine-grained shadow detection by exploiting shadow opacity characteristics in the scene. To this end, we propose a learning-based model that explicitly captures shadow position and opacity variations. In addition, we construct the first fine-grained shadow detection (FSD) dataset with opacity annotations across varied scenarios. Extensive qualitative and quantitative results show that our model can predict fine-grained shadow characteristics, achieving superior performance over the baselines and enhancing a wide range of applications, including shadow removal, shadow editing, and 3D reconstruction.

**Acknowledgments**

This work was partially supported by Guangdong Basic and Applied Basic Research Foundation (No.2022A1515110740), National Natural Science Foundation of China (No.62302356, No.62372360, No.62372352), CCF-ALIMAMA TECH Kangaroo Fund (NO.CCF-ALIMAMA OF 2025007), Youth Elite Scientist Sponsorship Program by China Association for Science and Technology (No.YESS20220610), and Innovation Capability Support Program of Shaanxi Province (No.2024ZC-KJXX-021).

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

# Appendix

This Appendix provides extended technical details and experimental analyses to support the main paper. We elaborate on the construction of the FSD dataset across diverse scenarios: indoor/outdoor environments, single/multiple light sources, shadow overlaps, and varying illumination intensities. We describe the physics-based rendering pipeline in Blender Cycles, leveraging path tracing to simulate realistic shadow interactions governed by radiometric principles. We elaborate in detail on the calculation process of the ground truth values for the opacity shadow mask and supplement the qualitative experimental results of shadow detection. Additionally, we further analyze and present the qualitative experimental results of shadow removal using the opacity shadow mask.

## A   FSD Dataset

### A.1   Additional details

When constructing the dataset, we considered a variety of scenarios encompassing indoor and outdoor environments, varying from single to multiple light sources, weak to strong lighting conditions, and situations with overlapping shadows. The dataset comprises shadow-free images, instance masks, binary shadow masks, self-shadow masks, and opacity shadow masks.

For indoor object models, we source 1030 models from Google Scanned Objects [7]. To prevent interference within the dataset, we manually exclude cases where a model consists of multiple separate parts. Human models are selected from free models on Sketchfab [21] based on quality, resulting in a manual selection of 59 models. In outdoor settings, we incorporate HDRI textures from Ambient CG [4] and Poly Haven. To ensure data quality, scenes with extremely weak lighting or existing shadows are excluded, leading to a final selection of 26 scenes. Below are some examples of scenes included in the dataset.

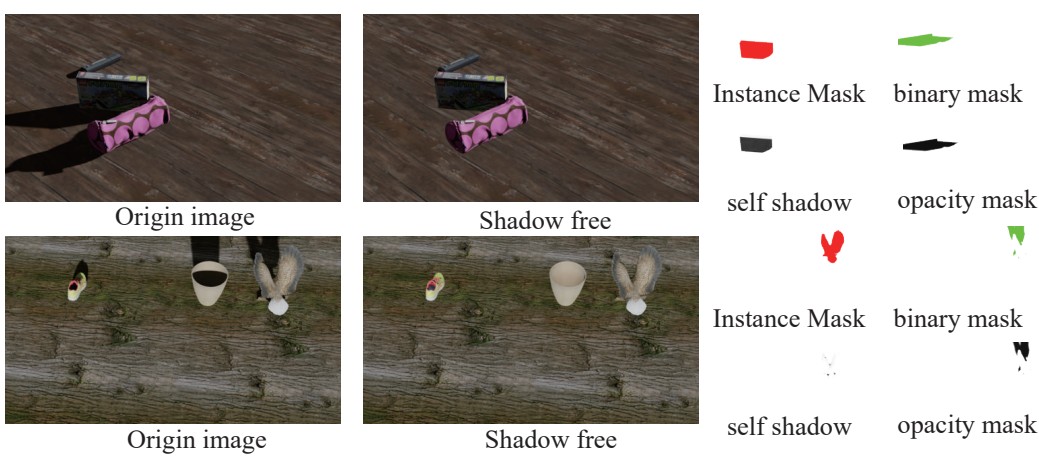

Figure 8: Indoor, single light

**Indoor, Single Light Scenario.**    In indoor settings, a cubic space serves as the base for constructing the scenes. A total of 428 floor textures from Poly Haven are utilized to adorn the indoor space. Subsequently, 1-5 objects are randomly positioned near the world coordinate origin, with their sizes randomly adjusted to diversify the dataset. The placement strategy for objects in this single-light indoor scenario (refer to Figure. 8) involves distributing the $xyxy$ coordinates within a narrow strip area to simulate common real-world object arrangements. For this scenario, a single light source (such as parallel light or SUN in Blender) is employed to replicate indoor reflection effects, aligning with typical indoor scene configurations. The variability in light sources encompasses factors like intensity, blur level (indicating the softness of shadows), light direction, position, and other parameters. Regarding the camera settings in both indoor and outdoor scenes, the position and angle of the camera are randomized, excluding specifications such as focal length and sensor size.

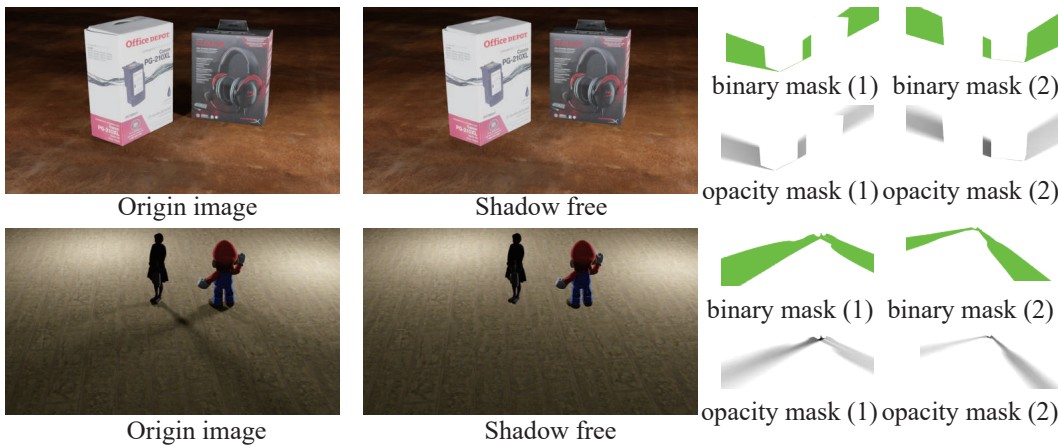

Figure 9: Indoor, multiple lights

**Indoor, multiple lights.**    In more intricate scenarios (refer to Figure. 9), the presence of multiple light sources may give rise to various shadow phenomena. This particular scenario has received limited attention in prior research. To enrich the dataset's scope, we have included this scenario to expand its comprehensiveness. In this setting, objects are placed with completely random xyxy coordinates. As for lighting, we have implemented a strategy to randomly generate lighting effects, encompassing point light (POINT in Blender), sunlight (parallel light, SUN), spotlight (SPOT), and area light (AREA). The selection of light types and quantities is randomized, and the attributes of the lights are determined in a similar random fashion as previously described.

**Shadow overlaps.**    When multiple light sources are utilized, shadow intersections occur (see Figure. 9), a phenomenon that cannot be adequately elucidated by a binary shadow mask alone. Our dataset intentionally includes scenarios featuring shadow intersections. The distinction between this and multiple-light scenarios lies in the placement strategy of objects and lights. Specifically, two objects are positioned to form a specific angle, after which two point light sources are placed in the directions of these objects (from the origin to each object) to generate intersecting shadows.

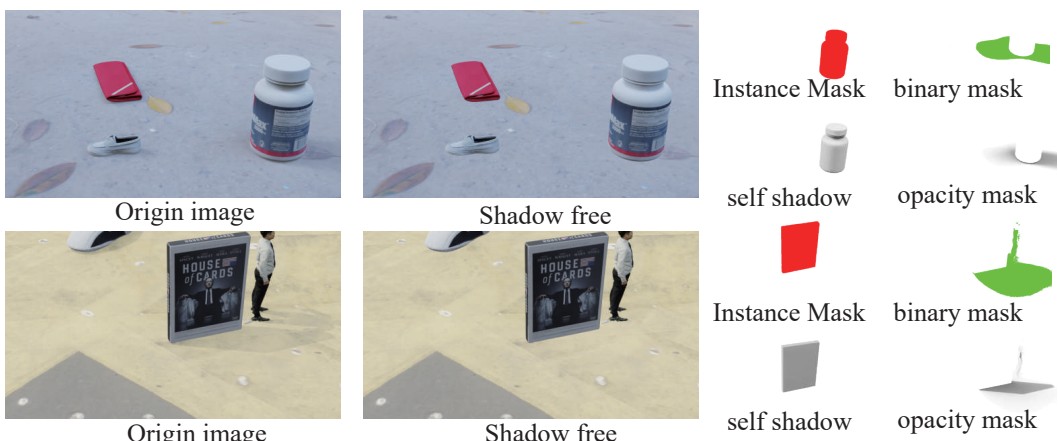

Figure 10: Outdoor, weak light

**Outdoor, weak light.**    For all outdoor scenes, we create a hemisphere and utilize HDRI textures to establish the environment, thereby integrating lighting details from the textures organically. In instances of low lighting intensity (refer to Figure. 10), shadows exhibit a blurred and ghosted appearance, rendering binary shadow masks less efficient for shadow elimination. Given that the lighting details are predetermined by the texture, the only elements subject to randomness are the camera angles and the placement of objects.

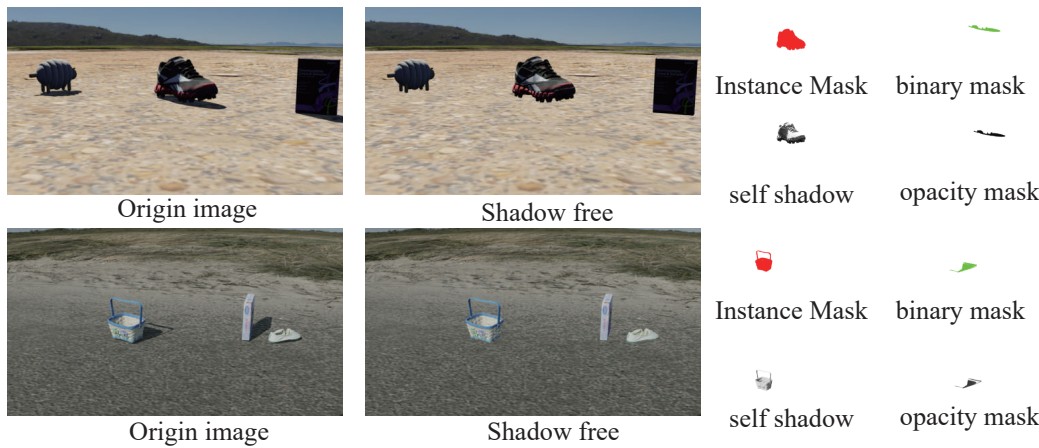
Figure 11: Outdoor, strong light

**Outdoor, strong light.** In outdoor environments characterized by intense lighting conditions (see Figure. 11), representative of real-world settings, shadows tend to be distinct and deep. Our dataset comprises a substantial quantity of such scenes. The primary contrast between these two outdoor setups lies in the HDRI textures employed.

Once the scene is set up, we employ keyframe animation to produce the initial image, a shadow-free rendition, and all desired annotations for each object. Blender sequentially generates the animation frame by frame, which is later subjected to post-processing. Due to inherent noise in the original opacity shadow mask from Blender Cycles, we begin by applying noise reduction techniques. Subsequently, we derive the binary shadow mask based on the opacity shadow mask. Following the processing of all samples, we compile a JSON file to document our datasets in the format specified by SOBA [44]. Comprehensive details regarding the scene setup will be released.

# B  Additional Shadow Detection Results

## B.1  Shadow Rendering Process

Blender Cycles uses the path tracing algorithm, which is based on the following equation:

$$L_o(p, \omega_o) = L_e(p, \omega_o) + \int_\Omega f_r(p, \omega_i, \omega_o) L_i(p, \omega_i) n \cdot \omega_i d\omega_i \tag{12}$$

It uses radiometry to describe lighting more accurately, resulting in more realistic images. The term $L_i(p, \omega_i) n \cdot \omega_i$ describes the radiance of the light coming from solid angle $\omega_i$ in the environment, contributing to the radiance at point $p$. Thus, $\int_\Omega L_i(p, \omega_i) n \cdot \omega_i d\omega_i$ represents the contribution of all light rays in the hemisphere $\Omega$, i.e. all light in the environment. The term $f_r$ determines how much light will be reflected. Although with numerous BSDF, BRDF models available, they only determine the material of the object. The determinant of shadows is the term $L_i$ .The term $L_e$ represents the object's self-emission.

The ground truth opacity shadow mask $\alpha_{gt}$ is calculated from the shadow image $S$ and the shadow-free image $F$. The specific steps are as follows: first, convert $S$ and $F$ to the YCbCr color space and extract the Y channel, then compute the ratio of $Y_S$ to $Y_F$, apply a low-pass filter to remove noise, and finally obtain $\alpha_{gt}$ through thresholding.

$$\alpha_{gt} = max(t, f(\frac{Y_S}{Y_F})) \tag{13}$$

Here, $Y_S$ and $Y_F$ represent the Y channels of the shadow image and the shadow-free image in the YCbCr color space, respectively. $f$ denotes a low-pass filtering operation with $\sigma = 0.5$, and $t$ is the threshold, which is set to 0.1 in this paper.

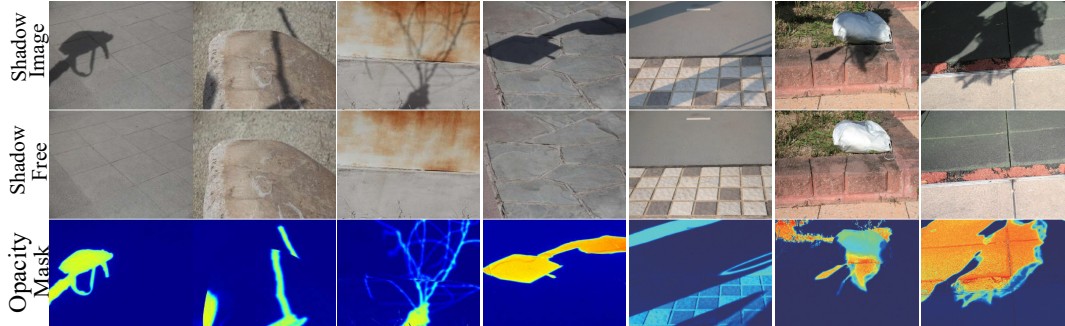

Figure 12: The figure demonstrates the shadow image, the shadow-free image, and the extracted opacity shadow mask. Compared to the binary mask, the opacity shadow mask not only indicates the location of the shadows but also reveals the intensity of the shadows and the degree of background.

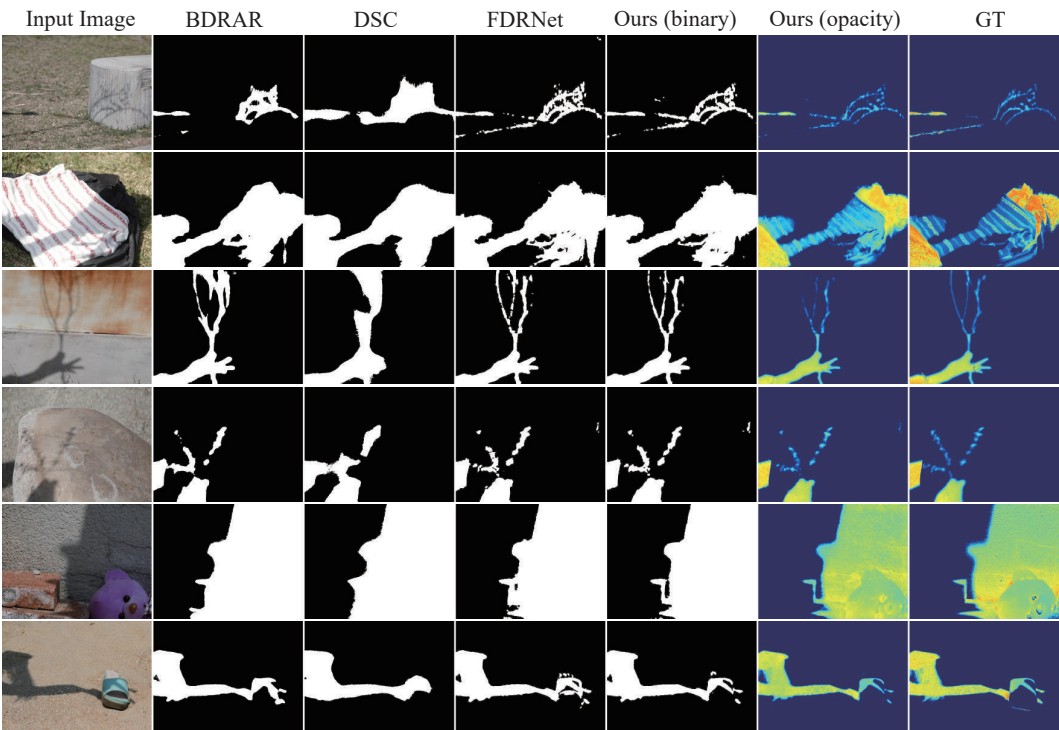

Figure 13: The figure illustrates qualitative comparison results between ours and the state-of-the-art methods on the SRD dataset [35], our method exhibits significant advantages in visual quality.

Figure 13 presents additional experimental results on the SRD dataset [35]. Our method outperforms the state-of-the-art approaches across multiple examples,our method demonstrates exceptional performance in fine-grained shadow detection. For instance, in rows 2 and 4, our method can predict the transparency changes of shadows, while in rows 1 and 5, it achieves more precise detection of soft shadow boundaries. These results further validate the reliability of the shadow detection approach guided by shadow opacity.

## B.2    Ablation Study

**Qualitative comparison.**    The ablation results in Figure 14 validate our design: the opacity mask enables robust shadow positioning and transparency estimation, complemented by $\mathcal{L}_{\mathrm{grad}}$ for regulating gradient smoothness and $\mathcal{L}_{\mathrm{area}}$ for boosting accuracy in extreme scenarios.

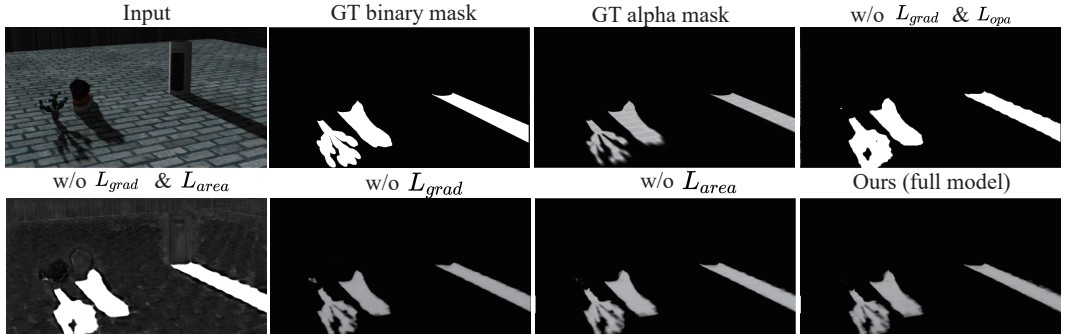

Figure 14: Qualitative comparison results of different ablated versions of our model.

**Quantitative Evaluation.** We additionally conducted an ablation study of our loss functions on the ISTD dataset [43]. As it lacks ground truth annotations for shadow opacity, we employed the Balanced Error Rate (BER) metric for evaluation. The results, presented in the table below, demonstrate that our full model outperforms all ablated versions,

| Method | BER ↓ | Shadow↓ | Non Shad.↓ |
|---|---|---|---|
| w/o $\mathcal{L}_{\text{grad}}$ and $\mathcal{L}_{\text{opa}}$ | 1.73 | **0.92** | 2.53 |
| w/o $\mathcal{L}_{\text{grad}}$ and $\mathcal{L}_{\text{area}}$ | 1.55 | 2.21 | 0.88 |
| w/o $\mathcal{L}_{\text{grad}}$ | 1.50 | 2.25 | **0.75** |
| w/o $\mathcal{L}_{\text{area}}$ | 1.46 | 2.15 | 0.77 |
| Ours (full) | **1.32** | 1.67 | 0.96 |

Table 7: Ablation study on ISTD.

performs all ablated versions, validating the effectiveness of our design choices.

## C    Additional Shadow Removal Results

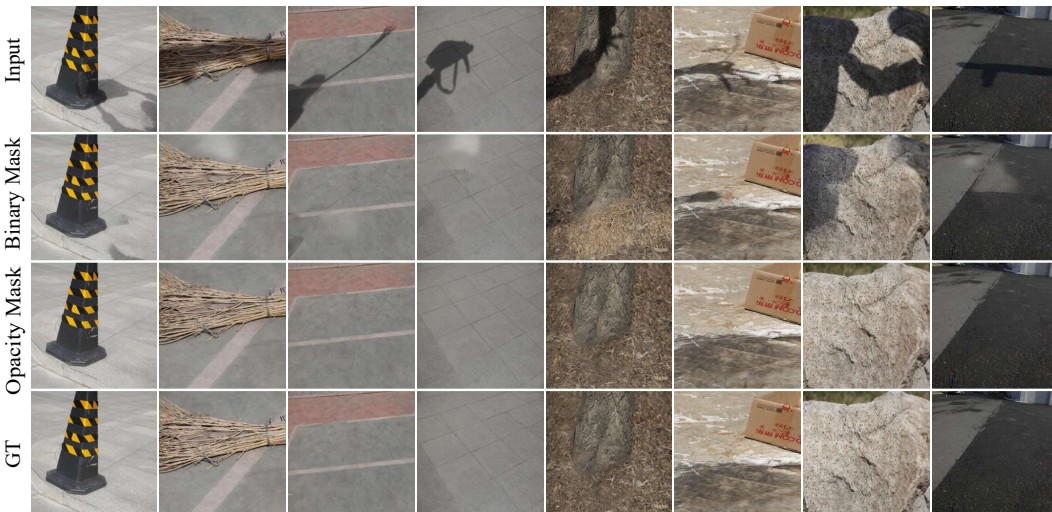

Figure 15: We compared the results of using binary masks and opacity masks for shadow removal in training the ShadowDiffusion[12] model on SRD Dataset [26]. The results indicate that using opacity masks instead of binary masks for annotation leads to better visual effects.

In our shadow removal experiments, we improved upon the state-of-the-art ShadowDiffusion [12] method. Specifically, we replaced the binary shadow mask with the detection results from our method for training, while keeping other hyperparameters and optimization strategies consistent with the original ShadowDiffusion. The experiments were conducted on an RTX 4090 GPU, with the training epochs set to 1000. We employed the Adam optimizer (with momentum parameters of (0.9, 0.999)) and an initial learning rate of $3 \times 10^{-5}$. The model weights were initialized using the Kaiming

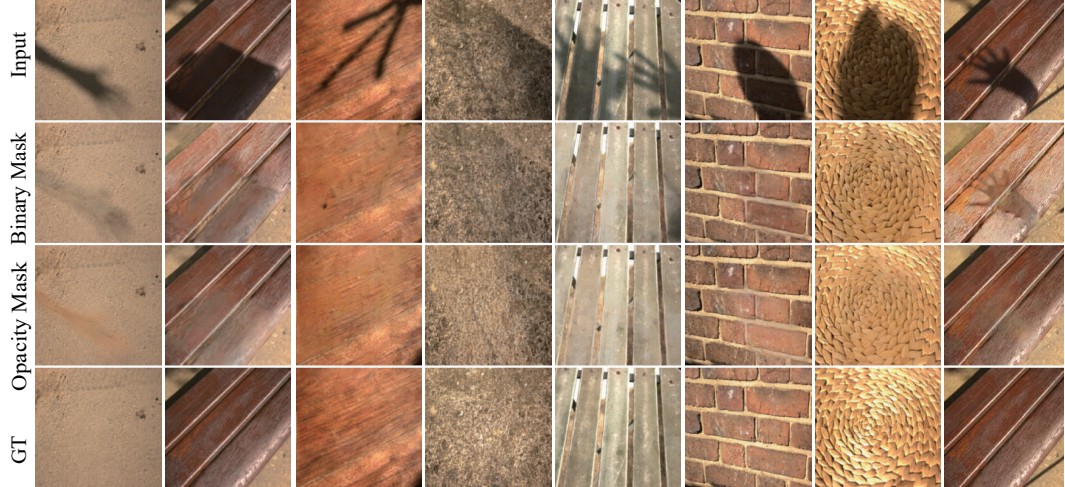

Figure 16: Results of different inputs on the LRSS [10] dataset using ShadowDiffusion [12].

initialization technique [15], and an exponential moving average (EMA) strategy with a decay rate of 0.9999 was applied across all experiments.

Figure 15 showcases the performance superiority of the model trained using opacity shadow masks compared to the model trained with binary shadow masks across various test images. The qualitative analysis demonstrates that, under the same training conditions, the utilization of opacity shadow masks enables the model to effectively eliminate shadows while maintaining more detailed information about the underlying scene. Particularly in complex scenarios with blurred shadow boundaries, the model trained with opacity shadow masks shows notably enhanced performance.

To validate the effectiveness of the opacity shadow mask, we conducted supplementary experiments on the LRSS soft shadow dataset [10], which contains 46 pairs of shadow and shadow-free images. The experiments adopted a strategy of training on the SRD dataset [26] and evaluating on the LRSS dataset [10]. As illustrated in Figure. 16, the opacity shadow mask demonstrated significant advantages in restoring soft shadow edges and background textures, thereby fully validating its effectiveness.

