# OpenReview forum: "Under the Shadow: Exploiting Opacity Variation for Fine-grained Shadow Detection"
_NeurIPS.cc/2025/Conference — NeurIPS 2025 poster_

### Official Review · Reviewer_WyPt · 2025-06-23

**Clarity:** 4
**Significance:** 2
**Originality:** 2
**Rating:** 5
**Confidence:** 3

**Summary:**

This paper proposes predicting an opacity map instead of a traditional binary shadow mask for the task of fine-grained shadow detection. To support this, the authors introduce a new dataset, FSD, and design three novel modules. The method achieves comparable or superior performance not only on the proposed dataset but also on existing shadow detection benchmarks, which is a notable strength. The paper is clearly written and easy to follow, and the extensive experimental results effectively demonstrate the efficacy of the proposed approach.

**Questions:**

I have several questions regarding the design choices made in the paper:

- The paper adopts a cumulative learning strategy, but it is not clear why this is necessary.
    - How crucial is this strategy to the final model performance?
    - Have the authors conducted any ablation or sensitivity analysis to validate its effectiveness?
    - How were the related hyperparameters selected?
- Can shadow removal models also be used for shadow detection?
    - For instance, given both the shadow and shadow-free images (as in the FSD dataset), one could compute an opacity map or binary mask from their difference.
    - Could such derived maps serve as a competitive baseline to the proposed method?
    - If so, is there a specific reason the authors chose not to compare against such approaches?
- The baseline comparisons seem outdated. According to Table 2, the most recent papers used for comparison are from 2021 (for ISTD) and 2020 (for FSD). This is a significant limitation. The authors should compare their method against more recent state-of-the-art works, especially dealing with soft shadows.
    - *HomoFormer: Homogenized Transformer for Image Shadow Removal*, CVPR 2024
    - *Detail-Preserving Latent Diffusion for Stable Shadow Removal*, CVPR 2025
    - *OmniSR: Shadow Removal under Direct and Indirect Lighting*, AAAI 2025
    - *DeS3: Attention-driven Self and Soft Shadow Removal using ViT Similarity and Color Convergence*, AAAI 2024

Please let me know if I have misunderstood any part of the paper.

**Ethical Concerns:**

["NO or VERY MINOR ethics concerns only"]

**Final Justification:**

After carefully revisiting both the authors’ rebuttal and the main paper, I acknowledge that the authors have addressed most of my concerns—particularly those regarding the ablation of the cumulative learning strategy and the use of straightforward or outdated baselines. Given the constraints of the rebuttal phase, where visual results cannot be included, the authors appear to have provided the best possible clarification.

**Limitations:**

Yes

**Paper Formatting Concerns:**

No concerns

**Quality:**

3

**Strengths And Weaknesses:**

**Strength**

- The paper is very well written, making it easy to understand both the shadow detection field and the specific problem addressed by the authors.
- The authors have made a clear effort to enhance the quality of the paper through comprehensive experimental results, comparative tables with existing datasets, and well-designed model architecture figures.
- The inclusion of a limitations section and an analysis of failure cases in the appendix contributes positively to the paper’s credibility and transparency.

**Weakness**

- Please refer to the Questions section below for specific concerns.

---

> ### Author Rebuttal · Authors · 2025-07-31
>
> Thanks for the thoughtful review and positive comments on the paper’s clarity, comprehensive experiments, and limitation discussions.
> Given the opportunity, we will incorporate all suggestions with details in the revision.
> Below, we address all concerns raised.
> We will be very happy to clarify further concerns (if any).
>
> **Q1: The paper adopts a cumulative learning strategy, but it is not clear why this is necessary...How were the related hyperparameters selected?**
>
> A1: Thanks for the valuable suggestion.
> The cumulative learning strategy (see L148–L151) dynamically balances position and opacity features by adjusting the fusion weight η over training epochs, optimizing fine-grained shadow detection accuracy.
> We validate the effectiveness of this strategy by conducting ablation studies on cumulative learning and hyperparameter selection in the FSD dataset, with the results shown below.
> We can see that the cumulative learning strategy is crucial for fine-grained shadow detection, and η = 0.5 achieves the best performance.
>
> | Method                     | BER (↓) | RMSE (↓) | WSE (↓) |
> |----------------------------|---------|----------|---------|
> | w/o cumulative learning     | 2.18    | 0.1505   | 10.83  |
> | η = 0.3                    | 1.93    | **0.0635**   | 6.21   |
> | η = 0.4                    | 1.83    | 0.9840    | 6.16   |
> | η = 0.6                    | 1.80    | 0.0675   | 6.17   |
> | **Ours (η = 0.5)**            | **1.78**| 0.0669| **5.78**|
>
> **Q2: Can shadow removal models also be used for shadow detection...Could such derived maps serve as a competitive baseline to the proposed method?**
>
> A2: Yes. Shadow removal models can also be used for shadow detection.
> However, we did not conduct such an experiment for two reasons.
> First, shadow removal is often considered as a downstream task of shadow detection, and thus is rarely used as an input for shadow detection.
> Second, even the SOTA shadow removal models cannot ensure the complete removal of all shadows in the image.
> Thus, using shadow removal models for shadow detection may contain a lot of noise with false positives and false negatives.
> To validate the above explanations, we conducted an experiment on the FSD dataset.
> HomoFormer, the shadow removal model, is used for shadow detection.
> The BER, RMSE, and WSE results are 5.31, 0.1071, and 8.32, respectively, which are significantly worse than our method (1.78, 0.0669, 5.78).
>
> **Q3: The baseline comparisons seem outdated...The authors should compare their method against more recent state-of-the-art works [1-4], especially dealing with soft shadows.**
>
> A3: Sorry for the confusion.
> Note that our motivation is that explicitly modeling the shadow opacity can help the downstream applications like shadow detection and shadow removal.
> Thus, the experimental comparisons are separated into two aspects, i.e., shadow detection and shadow removal.
>
> For shadow detection, as there is no existing method that predicts soft shadows explicitly, we compare with SOTA shadow detection methods in Table 2.
>
> For shadow removal, we have compared our method with SOTA shadow removal methods in Table 4, including HomoFormer[1] and DeS3[4].
> For OmniSR [3], as it involves additional inputs (i.e., Normal Map and Depth Map) in the training process with a new synthetic dataset, it would be difficult to compare the shadow removal performance directly.
> We further compare the results with Detail-Preserving Latent Diffusion (CVPR 2025) [2].
> On the SRD dataset with varied shadow opacities, the PSNR and SSIM results of [2] are 33.63 and 0.968, while our method achieves 34.84 and 0.974, respectively, confirming once again the effectiveness of our method on downstream applications like shadow removal.
> We will make the discussions more clearly, and conduct additional comparisons with recent SOTA methods in the revision.
>
>
> **References:**
>
> [1] Xiao et al., HomoFormer: Homogenized Transformer for Image Shadow Removal, CVPR 2024
>
> [2] Xu et al., Detail-Preserving Latent Diffusion for Stable Shadow Removal, CVPR 2025
>
> [3] Xu et al., OmniSR: Shadow Removal under Direct and Indirect Lighting, AAAI 2025
>
> [4] Jin et al., DeS3: Attention-driven Self and Soft Shadow Removal using ViT Similarity and Color Convergence, AAAI 2024

---

> > ### Comment · Reviewer_WyPt · 2025-08-05
> >
> > After carefully revisiting both the authors’ rebuttal and the main paper, I acknowledge that the authors have addressed most of my concerns—particularly those regarding the ablation of the cumulative learning strategy and the use of straightforward or outdated baselines. Given the constraints of the rebuttal phase, where visual results cannot be included, the authors appear to have provided the best possible clarification. Thank you for the thoughtful responses!
> >
> > **Additional Comments**
> >
> > - If I have understood correctly, the Q2 results presented in the rebuttal could be used to add straightforward soft shadow detection baseline results, utilizing state-of-the-art object removal models, to Table 2. Since it may seem trivial at first glance to obtain a soft shadow map when both shadow and shadow-free images are available, including such results and their analysis in the main paper would help strengthen the contribution and clarify the novelty.
> > - As noted in the paper, I encourage the authors to publicly release the code and dataset. This would enhance reproducibility and support future research in this area.

---

> > > ### Author Response · Authors · 2025-08-05
> > >
> > > We sincerely thank the reviewer for recognizing the contribution of our work, and appreciate your helpful suggestions.
> > > We are glad that our response has addressed the concerns.
> > >
> > > We will include a detailed discussion of the additional baselines, ablated versions, and other related works in the revised paper, and release both the code and dataset to facilitate future research in the community. Thank you again!

---

### Official Review · Reviewer_cS6y · 2025-07-02

**Clarity:** 2
**Significance:** 2
**Originality:** 2
**Rating:** 4
**Confidence:** 4

**Summary:**

This paper presents a fine-grained shadow detection method that uses opacity instead of binary masks. It models shadows as having varying degrees of opacity rather than treating them as uniformly opaque regions, better reflecting the physical interactions of light with objects and surfaces. The proposed method consists of three components: a Shadow Opacity Augmentation (SOA) module to simulate varied shadow opacities during training, a Shadow Feature Separation (SFS) module to disentangle shadow position and opacity features, and an Opacity Mask Prediction (OMP) module to generate a continuous shadow opacity map. Furthermore, the paper introduces a new dataset — Fine-grained Shadow Detection (FSD) — containing opacity-annotated masks under varied real-world conditions. Extensive experiments, including comparisons and ablations, show superior performance in shadow detection and downstream tasks (e.g., shadow removal and editing), demonstrating the effectiveness of opacity-aware modeling.

**Questions:**

Please see Weaknesses.

**Ethical Concerns:**

["Major Concern: Data quality and representativeness"]

**Final Justification:**

Most of my concerns have been addressed well in the rebuttal. As I wrote to the authors, the limitations and the additional results should definitely be included in the revised paper.

**Limitations:**

Yes.

**Paper Formatting Concerns:**

None.

**Quality:**

2

**Strengths And Weaknesses:**

Strengths:

* The paper is clearly written. The module explanations and illustrations are easy to follow.

* The method is well-motivated, and each module in the pipeline is designed for effective learning. The experiments, including comparisons with state-of-the-art methods, ablation studies, and downstream application demonstrations, are relatively comprehensive.

* Using opacity instead of binary masks for fine-grained shadow representation demonstrates effectiveness in shadow detection research and related tasks such as image editing, relighting, and scene understanding.

* The concept of modeling shadow opacity variations and explicitly learning opacity-aware representations seem novel in the context of shadow detection. The construction of the FSD dataset adds further value, as it addresses an important gap in existing resources.


Weaknesses:

* There is a lack of direct and comprehensive comparison with similar continuous mask representations. While the paper compares its approach against state-of-the-art binary mask detectors, it does not explore baselines from related fields such as alpha matting and soft segmentation, which also produce continuous-valued outputs. Additionally, some prior work has explored similar representations for example using soft masks [1]. Including discussions and comparisons with these methods could strengthen the evaluation.

* More experimental results are needed to demonstrate the method's effectiveness. Specifically:

(a) Opacity estimation in complex scenes. While briefly acknowledged, model limitations in handling multiple interacting shadows or varying illumination directions warrant deeper analysis.

(b) Comparisons with recent shadow removal methods. More thorough comparisons and discussions with recent works [1–4] would provide a clearer picture of the method’s strengths and weaknesses in shadow removal and downstream tasks.

* Evaluation limitations. The proposed Weighted Shadow Error (WSE) metric seems promising, but its interpretation and correlation with downstream task quality could be elaborated further. There is limited discussion on how this new metric aligns with human perception or real-world usability.

* Potential dataset bias.  It is unclear how diverse the newly introduced FSD dataset is in terms of scene diversity, lighting conditions (e.g., artificial, natural, mixed), or camera types. Limited diversity in these areas could hinder the generalization ability of models trained on this dataset. It would be helpful to include a characterization map or summary table highlighting key dataset attributes such as: Distribution of indoor vs. outdoor scenes; Proportions of different lighting types (e.g., daylight, incandescent, LED); Variety of camera sensors or sources.


[1] Wang et al. SoftShadow: Leveraging Soft Masks for Penumbra-Aware Shadow Removal. CVPR 2025.

[2] Wang et al. MetaShadow: Object-Centered Shadow Detection, Removal, and Synthesis. CVPR 2025.

[3] Lu et al. EvenFormer: Dynamic Even Transformer for Real-World Image Restoration. CVPRW 2025.

[4] Fu et al. Multi-level Feature Fusion Network for Shadow Removal Detection. TCSVT 2025.

---

> ### Author Rebuttal · Authors · 2025-07-31
>
> We thank the reviewer for recognizing the presentation, motivation, and the extensive evaluation of our work.
> Given the opportunity, we will incorporate all suggestions with details in the revision.
> Below, we address all concerns raised.
> We will be very happy to clarify further concerns (if any).
>
> **Q1: There is a lack of direct and comprehensive comparison with similar continuous mask representations...explore baselines from related fields such as alpha matting and soft segmentation, which also produce continuous-valued outputs.**
>
> A1: Thanks for the valuable suggestion.
> We follow the continuous mask creation process from related fields such as alpha matting and soft segmentation, and conduct additional comparisons as follows.
> In particular, the baseline combines Alpha Matting [5] with the binary shadow mask detection method [6] to produce continuous-valued outputs as the estimated shadow opacity.
> The BER, RMSE, and WSE results of the above baseline on the FSD dataset are 3.01, 0.1405, and 8.05, respectively, while our method achieves 1.78, 0.0669, and 5.78, respectively.
> These results demonstrate that, unlike our method, existing general continuous mask representations do not consider the physical properties of shadows, leading to poor performance in estimating shadow opacity.
>
> **Q2: Some prior work has explored similar representations, for example, using soft masks [1]. Including discussions and comparisons with these methods could strengthen the evaluation.**
>
> A2: As stated in L80-L82, we model the shadow opacity variations explicitly by the underlying physics of light for shadow detection, while SoftMask [1] generates soft masks as intermediate results for shadow removal.
> Thus, SoftMask [1] cannot be easily adapted to downstream applications like shadow detection and editing.
> We will add a thorough discussion of these existing works in the revision.
>
> **Q3: Opacity estimation in complex scenes. While briefly acknowledged, model limitations in handling multiple interacting shadows or varying illumination directions warrant deeper analysis.**
>
> A3: Due to the page length limitation, we discuss the limitations briefly in the paper (see L271-L275), and show detailed visual analysis in the supplemental (see Fig. 11 and Sec. D).
> In scenarios like forests or low-light environments with scattered or overlapping shadows, while the performance of our method’s opacity estimation drops, existing binary shadow representations (without continuous value gradients) often perform even worse due to the lack of smooth transitions.
> One possible solution is to consider scene illumination properties for shadow opacity estimation in complex scenes.
>
> **Q4: Comparisons with recent shadow removal methods. More thorough comparisons and discussions with recent works [1–4].**
>
> A4: As these works were published recently after the NeurIPS submission deadline, they are not open-sourced yet.
> We consider these shadow removal methods as concurrent works, and will add detailed analysis in the revision.
>
> As stated in L80-L82, SoftMask [1] generates soft masks as intermediate results for shadow removal, while we model the shadow opacity variations explicitly by the underlying physics of light for shadow detection.
> Thus, it cannot be easily adapted to downstream applications like shadow detection and editing.
>
> MetaShadow [2] focuses on instance-level shadow detection, removal, and synthesis, while we focus on all shadows in the scene.
> Instance-level fine-grained shadow detection may help the downstream applications like image editing, and could be an interesting future work.
>
> EvenFormer [3] is specifically designed for real-world image restoration, and all experiments are based on different scenarios (i.e., NTIRE25-Shadow-Test dataset).
>
> The goal of Fu et al. [4] is to detect the remaining shadows in the image after the shadow removal process, rather than shadow removal.
>
> **Q5: Evaluation limitations. The proposed Weighted Shadow Error (WSE) metric seems promising...there is limited discussion on how this new metric aligns with human perception or real-world usability.**
>
> A5: BER and RMSE are commonly used to measure the accuracy of discrete and continuous labels, respectively (see L191-L194).
> Although RMSE can also be used for our fine-grained shadow detection problem, it does not distinguish shadow and non-shadow regions in the image.
> As non-shadow regions dominate in most images, we use the weighted strategy to pay more attention to nuanced shadow properties in shadow regions.
> Thus, we propose a new metric, named Weighted Shadow Error (WSE), which is a weighted RMSE value for shadow detection between the shadow and non-shadow regions.
>
> To further validate whether the proposed WSE metric better aligns with human perception, we conduct a user study below.
> For a set of images, we consider the users' average ranking order $seq_{user}$ of fine-grained shadow detection quality for these images, and measure the similarity between $seq_{user}$ and the ranking order $seq_{metric}$ derived by the evaluation metric.
> A higher similarity score represents that this metric aligns better with human perception of shadow detection quality.
> We recruit 20 participants and randomly select 10 images from our testing dataset.
> In each comparison, 3 detected results by our method are displayed in random order, and the corresponding input and ground truth are also displayed side by side.
> The participants are asked to give the ranking order based on the shadow detection quality.
> In the end, the similarity score of WSE is 83% and the similarity score of RMSE is 64%, confirming once again that the new WSE metric aligns better with human perception of shadow detection quality.
>
> **Q6: Potential dataset bias. It is unclear how diverse the newly introduced FSD dataset is in terms of scene diversity, lighting conditions, or camera types...**
>
> A6: We construct the FSD dataset with several guidelines, including object, scene, and light source properties (see L102-l112 in the paper).
> The detailed construction process is illustrated in the supplemental (see L12-L65).
> We will visualize and highlight the key dataset attributes more clearly in the revision.
>
> In particular, the FSD dataset ensures diversity through the following guidelines to avoid dataset bias:
> + Scenes: The number of indoor and outdoor scenes is 1113 and 514, respectively. The number of objects in each scene ranges from 1 to 5.
> + Lighting: Brightness and positions are randomly sampled.
> + Camera: All cameras have a 50mm focal length, AUTO sensor setting, and randomized positions.
>
> **References:**
>
> [1] Wang et al., SoftShadow: Leveraging Soft Masks for Penumbra-Aware Shadow Removal. CVPR 2025.
>
> [2] Wang et al., MetaShadow: Object-Centered Shadow Detection, Removal, and Synthesis. CVPR 2025.
>
> [3] Lu et al., EvenFormer: Dynamic Even Transformer for Real-World Image Restoration. CVPRW 2025.
>
> [4] Fu et al., Multi-level Feature Fusion Network for Shadow Removal Detection. TCSVT 2025.
>
> [5] Gastal et al., Shared sampling for real‐time alpha matting. CGF 2010.
>
> [6] Zhu, et al., Mitigating intensity bias in shadow detection via feature decomposition and reweighting. ICCV, 2021.

---

> > ### Comment · Reviewer_cS6y · 2025-08-07
> >
> > Dear Authors,
> >
> > Thanks for the rebuttal. For Q1, I would really like to see more results in the revised paper. For Q3, it is important to have this discussion of limitation in the paper as well. I still don't think it is fair to claim this is a general dataset (Q6) but of course that is a gray area.
> >
> > The other concerns are addressed well. I am happy to increase my score to borderline accept. I still think the paper would benefit from a revision, which hopefully can be finalized till the deadline.
> >
> > Best.

---

> > > ### Author Response · Authors · 2025-08-08
> > >
> > > Thanks for your kind suggestions. We appreciate your willingness to accept our submission.
> > >
> > > In the revised paper, to thoroughly validate the effectiveness of our method, we will add more results and discussions for the shadow detection and downstream applications, and add more limitation discussions. As we make the first attempt to explore the new shadow detection task, we will revise the dataset presentation to avoid generalization confusion, and release both the dataset and code to facilitate future research in this direction.
> > >
> > > The suggested references, additional evaluations, and further discussion will be included in the final version of our paper. Thank you again!

---

### Official Review · Reviewer_DKcE · 2025-07-03

**Clarity:** 3
**Significance:** 2
**Originality:** 3
**Rating:** 4
**Confidence:** 3

**Summary:**

The authors propose a new shadow detection task and corresponding method to solve it - fine-grained shadow detection. Whereas existing methods mainly predict binary shadow masks, this task consists of creating an opacity mask for every pixel in an input image, which allows opacity variation in shadows (due to full vs partial occlusion) to be captured. The authors argue that this finer-grained detection can improve performance for downstream tasks, such as shadow removal and editing. The proposed method leverages a simulated dataset proposed by the authors and trains a network to learn opacity masks from an input image and augmentations of the GT shadow masks (during training). The authors report results on both shadow detection and downstream applications of shadow removal and editing.

**Questions:**

My main concerns are weaknesses (1) and (2) above, informing my current recommendation.

In addition, a few other smaller comments:
- You may consider adding other tasks to line 22 and the introduction - in particular 3D reconstruction from shadows [1] [2], since, at least to me, it seems much more clear that this application would benefit from fine-grained shadow detection (since weaker shadows are a function of partial occlusion, thus reflecting information about the underlying scene geometry). This could partially address my first concern. Getting results for 3D reconstruction from shadows that show more substantive improvements would address my concern further, but I acknowledge this may be challenging in the timeline of the rebuttal.
- Organization: I’d suggest moving the dataset section to before the experiments
- Further motivation why the new metric (weighted shadow error) is needed should be added
- Should equation 5 have an absolute value (i.e. L1 loss)?
- L188 typo: covert —> convert

[1] Towards Learning Neural Representations from Shadows. https://doi.org/10.1007/978-3-031-19827-4_18
[2] What You Can Reconstruct from a Shadow. https://doi.org/10.1109/CVPR52729.2023.01636

**Ethical Concerns:**

["NO or VERY MINOR ethics concerns only"]

**Final Justification:**

While I still have some doubts about the significance of impact on downstream tasks, based on the strong rebuttal, I'd like to raise my score to borderline accept since major concerns about motivation and missing comparisons/ablations have been mitigated.

**Limitations:**

yes

**Quality:**

3

**Strengths And Weaknesses:**

**Strengths**
- The proposed task is novel - and, in general, I believe that modeling more of the underlying physics of light, as attempted in this work, can be beneficial for computer vision pipelines. Moreover, the proposed solution is simple yet effective.
- Shadow detection results support the efficacy of the method.

**Weaknesses**
- The results for downstream applications show marginal improvements (Table 4). In addition, providing intuition for why “a fine-grained formulation of shadow regions” is “vital for scene understanding” (L36) would significantly improve the motivation for the paper. This notion is vaguely referred to again on L41 (...“could offer useful scene contextual clues”), but making this more concrete would be helpful. See my notes below - 3D reconstruction from shadows might be one more intuitive example for why fine-grained shadows can be helpful.
- I'd suggest the authors provide a baseline of the “trivial” solution described in 4.1 to substantiate the claim that using an auto encoder is less effective than the proposed technique. In addition, ablating the impact of the shadow augmentation in training would be helpful since this seems to be one of the main technical contributions.
- The authors fail to mention in the main text that the proposed dataset is simulated. Moreover, L106 is misleading as it states the dataset is “captured”, when, in fact, from my understanding based on the supplement, it is actually rendered.

---

> ### Author Rebuttal · Authors · 2025-07-31
>
> Thanks a lot for your detailed review and insightful suggestions.
> We are encouraged by your recognition of the novelty of our new problem and the effectiveness of our approach.
> Given the opportunity, we will incorporate all suggestions with details in the revision.
> Below, we address all concerns raised.
> We will be very happy to clarify further concerns (if any).
>
> **Q1: The results for downstream applications show marginal improvements (Table 4).**
>
> A1: The results in the "All" region are obtained by combining shadow and non-shadow areas, where non-shadow regions dominate in most images, reducing the visibility of shadow-specific improvements.
> As our method is specifically designed to handle the shadow opacity, we can see that our method achieves higher performance improvement for the shadow region in Table 4.
>
> Additionally, our opacity masks achieve consistent performance (best or second best) across datasets (ISTD, SRD), unlike existing methods like ShadowDiffusion or HomoFormer, in which the performance varies a lot due to dataset-specific shadow characteristics.
>
> **Q2: Providing intuition for why “a fine-grained formulation of shadow regions” is “vital for scene understanding” (L36) would significantly improve the motivation for the paper...3D reconstruction from shadows might be one more intuitive example...**
>
> A2: We totally agree with the reviewer that additional applications like 3D reconstruction from shadows can significantly improve our motivation. (see discussions with [2] in L19-L24).
> Fine-grained shadow properties, such as penumbra width and opacity variations, provide cues about occluder distance and light source intensity, which are critical for scene geometry understanding.
>
> As suggested, we conduct additional experiments for the 3D reconstruction task.
> Take the Bunny scene as an example.
> We retrain the 3D reconstruction model [1] with the shadow regions predicted by either FDRNet (binary mask) or our method (continuous opacity mask).
> Following the evaluation protocol in [1], we quantitatively analyze the quality of the reconstructed meshes by different methods.
> The RMSE value of FDRNet between the predicted point cloud and the ground truth point cloud is 0.01206, while the RMSE value of our method is 0.00928.
> These results further demonstrate the effectiveness of our method in enhancing scene understanding capabilities.
>
> Note that [1] claims that their method assumes a binary label on shadows and does not consider soft shadows (in the Discussions Section) yet.
> The initial experiment we conducted above also proves such an idea that utilizing fine-grained shadow properties could enhance the 3D reconstruction performance.
> Due to the limited time of the rebuttal, we will add thorough discussions for 3D reconstruction in the revised introduction and experiment sections.
>
> **Q3: Provide a baseline of the “trivial” solution described in 4.1 to substantiate the claim that using an auto encoder...ablating the impact of the shadow augmentation in training...**
>
> A3: Thanks for the valuable suggestions.
> Two additional quantitative comparisons on the FSD dataset are added to the table below.
> One is a baseline with an auto-encoder structure (L128-L129), and the other is an ablated version without using the shadow augmentation strategy.
> We can see that both strategies, i.e., a trivial solution like an auto-encoder and an ablated version without shadow augmentation, cannot achieve good performance, validating the effectiveness of our model design.
>
> | Method                     | BER (↓) | RMSE (↓) | WSE (↓) |
> |----------------------------|---------|----------|---------|
> | AutoEncoder            | 4.01    | 0.0919   | 10.49   |
> | Ours (w/o Augmentation)    | 2.16    | 0.0744   | 6.02    |
> | **Ours (full)**            | **1.78**| **0.0669**| **5.78**|
>
>
> **Q4: The authors fail to mention in the main text that the proposed dataset is simulated...**
>
> A4: We apologize for our tone and will make it clear that we use a physics-based simulator to ensure precise opacity annotations.
> Due to the page length limitation, we just highlight the main difference between our dataset and the existing dataset in the paper (see Table 1), and provide a detailed dataset construction process in the supplementary (see Sec. A).
> As we make the first attempt towards the fine-grained shadow detection problem, we believe that constructing a real shadow dataset with varied shadow properties can be an interesting and useful future work.
> Our dataset and code will also be released to facilitate future research in the community.
>
> **Q5: Organization: I’d suggest moving the dataset section to before the experiments.**
>
> A5: As suggested, we will reorganize the paper to place the dataset section before the experiments.
>
> **Q6: Further motivation why the new metric (weighted shadow error) is needed should be added.**
>
> A6: BER and RMSE are commonly used to measure the accuracy of discrete and continuous labels, respectively (see L191-L194).
> Although RMSE can also be used for our fine-grained shadow detection problem, it does not distinguish shadow and non-shadow regions in the image.
> As non-shadow regions dominate in most images, we use the weighted strategy to pay more attention to nuanced shadow properties in shadow regions.
> Thus, we propose a new metric, named Weighted Shadow Error (WSE), which is a weighted RMSE value for shadow detection between the shadow and non-shadow regions.
>
> To further validate whether the proposed WSE metric better aligns with human perception, we conduct a user study below.
> For a set of images, we consider the users' average ranking order $seq_{user}$ of fine-grained shadow detection quality for these images, and measure the similarity between $seq_{user}$ and the ranking order $seq_{metric}$ derived by the evaluation metric.
> A higher similarity score represents that this metric aligns better with human perception of shadow detection quality.
> We recruit 20 participants and randomly select 10 images from our testing dataset.
> In each comparison, 3 detected results by our method are displayed in random order, and the corresponding input and ground truth are also displayed side by side.
> The participants are asked to give the ranking order based on the shadow detection quality.
> In the end, the similarity score of WSE is 83% and the similarity score of RMSE is 64%, confirming once again that the new WSE metric aligns better with human perception of shadow detection quality.
>
> **Q7: Should equation 5 have an absolute value? L188 typo: covert to convert.**
>
> A7: Sorry for the typos. We indeed use the absolute value in Eq. 5. We will proofread and correct all typos in the revised paper.
>
> **References:**
>
> [1] Tiwary et al., Towards learning neural representations from shadows, ECCV 2022.
>
> [2] Liu et al., What You Can Reconstruct from a Shadow, ICCV 2023.

---

> > ### Comment · Reviewer_DKcE · 2025-08-05
> >
> > I'd like to thank the reviewers for thoroughly addressing my comments and concerns.
> >
> > It was good to see the new result for 3D reconstruction from binary vs the proposed method's predicted continuous shadow maps - this application is compelling. In the next revision, I'd suggest the authors consider re-running the 3D reconstruction a couple more times with different seeds to establish statistical significance.
> >
> > The new comparisons with the "trivial solution" (a vanilla autoencoder) and the proposed method w/o augmentations are both helpful and indicate that the proposed method yields a significant boost in performance. In addition, Reviewer cS6y raised a compelling point about other missing baselines, which I am happy to see the authors have since added during the rebuttal. In the next revision, I recommend the authors make it clear that the dataset is simulated in the main text (this can be done without taking much space, e.g. just add the word "simulated").
> >
> > While I still have some doubts about the significance of impact on downstream tasks, based on the strong rebuttal, I'd like to raise my score to borderline accept.

---

> > > ### Author Response · Authors · 2025-08-06
> > >
> > > Thank you very much for your thorough feedback and for raising the score to borderline accept. We sincerely appreciate your recognition of the contributions of our work, as well as your constructive suggestions.
> > >
> > > As suggested, we will run the 3D reconstruction task thoroughly to ensure statistical significance, and carefully incorporate all suggested improvements (including the dataset descriptions) into the revised paper.
> > >
> > > As we make the first attempt to explore the new shadow detection task, both the dataset and code will be released to facilitate future research in the community.

---

### Official Review · Reviewer_Xc6q · 2025-07-08

**Clarity:** 3
**Significance:** 3
**Originality:** 3
**Rating:** 4
**Confidence:** 3

**Summary:**

This paper introduces a novel approach to fine-grained shadow detection by explicitly modeling opacity variations in shadow regions. To train the model, the authors propose the first dataset with pixel-wise shadow opacity annotations across diverse scenarios. Experiments demonstrate that the proposed method can improve the shadow detection accuracy and benefit shadow removal.

**Questions:**

Since the shadow mask is not the final output, I would like to see more visual comparisons on downstream tasks such as shadow removal. In addition, the paper does not include experiments evaluating the impact of the $L_{opa}$ and $L_{pos}$ losses in the Shadow Feature Separation module.

**Ethical Concerns:**

["NO or VERY MINOR ethics concerns only"]

**Final Justification:**

The authors' rebuttal has solved most of my concerns, and I will keep my rating as Borderline Accept.

**Limitations:**

Yes

**Paper Formatting Concerns:**

No major concerns.

**Quality:**

3

**Strengths And Weaknesses:**

Strengths:
- 1. This paper is well-written and easy to follow.
- 2. This paper introduces the first dataset with opacity-annotated shadow masks, filling the gap in the field.

Weakness:
- 1. Since shadow removal is the primary downstream application of shadow detection, more visual comparisons should be provided between the fine-tuned ShadowDiffusion model using the proposed method and other shadow removal methods using alternative shadow detection approaches.
- 2. In Table 4, the improvement in the "All" region appears relatively minor. An explanation for this would be helpful.
- 3. In Table 3, the authors should clarify which dataset was used for testing. I hope to know each design's performance on ISTD and FSD dataset.
 - 4. Ablation study for the $L_{opa}$ and $L_{pos}$ should be provided.

---

> ### Author Rebuttal · Authors · 2025-07-31
>
> Thanks a lot for your time and constructive feedback.
> We are encouraged by your recognition of our motivation with good presentation and contribution.
> Given the opportunity, we will incorporate all suggestions with details in the revision.
> Below, we address all concerns raised.
> We will be very happy to clarify further concerns (if any).
>
> **Q1: More visual comparisons should be provided between the fine-tuned ShadowDiffusion model using the proposed method and other shadow removal methods using alternative shadow detection approaches.**
>
> A1: As suggested, we conduct more visual comparisons between different methods (ShadowDiffusion + Our Mask, ShadowDiffusion + DHAN, ShadowDiffusion + FDRNet, HomoFormer + FDRNet).
> However, since visual comparison is unavailable in the rebuttal, we also conduct a user study to show visual quality comparisons from the human perception perspective.
> We will show and discuss these visual comparisons in the paper and the supplementary material.
>
> For the user study, 20 participants are involved, and 10 images are randomly sampled from the SRD dataset.
> Each time, participants are shown an input shadow image with 4 shadow removal results by the above methods.
> Participants are asked to vote for the best shadow removal result.
>
> In the end, the results of ShadowDiffusion + Our Mask are ranked the best in 55% of the votes, HomoFormer + FDRNet in 25% of the votes, ShadowDiffusion + FDRNet in 10% of the votes, and ShadowDiffusion + DHAN in 10% of the votes, which confirms the superior visual quality of our approach.
>
> **Q2: In Table 4, the improvement in the "All" region appears relatively minor. An explanation for this would be helpful.**
>
> A2: We appreciate the valuable suggestion and will add detailed discussions in the revised paper.
>
> The results in the "All" region are obtained by combining shadow and non-shadow areas, where non-shadow regions dominate in most images, reducing the visibility of shadow-specific improvements.
> As our method is specifically designed to handle the shadow opacity, we can see that our method achieves higher performance improvement for the shadow region in Table 4.
>
> Additionally, our opacity masks achieve consistent performance (best or second best) across datasets (ISTD, SRD), unlike existing methods like ShadowDiffusion or HomoFormer, in which the performance varies a lot due to dataset-specific shadow characteristics. The user study results mentioned in Q1 further demonstrate the effectiveness of our method in improving shadow removal visual quality.
>
> **Q3: In Table 3, the authors should clarify which dataset was used for testing. I hope to know each design's performance on ISTD and FSD dataset.**
>
> A3: Sorry for the confusion.
> We use the FSD dataset for testing in Table 3.
> Since ISTD does not provide the ground truth for the shadow opacity, we use the BER metric for evaluation.
> As shown in the table below, our full model outperforms ablated versions, validating the effectiveness of our design choices.
>
> | Method                                | BER ↓ | Shadow ↓ | Non Shad. ↓ |
> |---------------------------------------|-------|----------|-------------|
> | w/o opacity mask                      | 1.73  | **0.92**     | 2.53        |
> | w/o $L_{\text{grad}}$ and $L_{\text{area}}$ | 1.55  | 2.21     | 0.88        |
> | w/o $L_{\text{grad}}$       | 1.50  | 2.25     | **0.75**        |
> | w/o $L_{\text{area}}$       | 1.46  | 2.15     | 0.77        |
> | **Ours (full)**                       | **1.32** | 1.67 | 0.96    |
>
> **Q4: Ablation study for the $L_{pos}$ and $L_{opa}$ should be provided.**
>
> A4: As suggested, we conduct additional ablation studies for $L_{pos}$ and $L_{opa}$ on FSD as follows.
> For the BER metric, the results of w/o $L_{opa}$, w/o $L_{pos}$, and ours (full) are 1.97, 2.82, and 1.78, respectively.
> For the WSE metric, the results of w/o $L_{pos}$, and ours (full) are 5.93 and 5.78, respectively.
> These results confirm that both losses are critical for fine-grained shadow detection, with $L_{opa}$ enhancing shadow opacity modeling and $L_{pos}$ ensuring accurate shadow position.
>
> **Q5: I would like to see more visual comparisons on downstream tasks such as shadow removal.**
>
> A5: As mentioned in Q1, we will add more visual comparisons for downstream tasks in the revised paper.
> In addition, to further demonstrate the usefulness of our opacity masks in downstream applications, we explore another task, i.e., 3D reconstruction, which is suggested by Reviewer DKcE.
>
> We retrain the 3D reconstruction model [1] with the shadow regions predicted by either FDRNet (binary mask) or our method (continuous opacity mask).
> Following the evaluation protocol in [1], we quantitatively analyze the quality of the reconstructed meshes by different methods.
> Take the Bunny scene as an example.
> The RMSE value with FDRNet between the predicted point cloud and the ground truth point cloud is 0.01206, while the RMSE value with our opacity mask is 0.00928.
> These results further demonstrate the effectiveness of our method in enhancing scene understanding capabilities.
>
> **References:**
>
> [1] Tiwary et al., Towards learning neural representations from shadows, ECCV 2022.

---

### Author Response · Authors · 2025-08-05

Dear Reviewers,

We greatly appreciate your time and effort in reviewing this paper.

As the discussion period is nearing its end, we just want to ensure that we can address all raised concerns thoroughly.
If you have any remaining questions or concerns that you'd like us to clarify, please share them with us so that we can address them before the discussion deadline.

Thanks again for your insightful and valuable feedback.

---

### Comment · Area_Chair_BjRf · 2025-08-05

Dear Reviewers,

As we approach the end of the author–reviewer discussion phase (August 8, 11:59pm AoE), I kindly remind you to review the authors’ rebuttals carefully, particularly the sections that address your specific comments.

Please consider whether the authors’ responses adequately address your concerns. If not, feel free to engage in further discussion while the window is still open. Kindly note that submitting a “Mandatory Acknowledgement” without posting any discussion or comments to the authors is not permitted.

Your timely participation is essential to ensure a fair and constructive review process. If you feel your concerns have been sufficiently addressed, you may also proceed to submit your Final Justification and update your rating ahead of the deadline.

Thank you again for your valuable contributions.

Best regards,\
AC

---

### Note · Authors · 2025-08-13

We thank the AC and reviewers for their time and effort during the reviewing and discussion processes.
We've carefully considered each comment and provided detailed responses to all concerns in the rebuttal.

We are glad that ALL Four Reviewers expressed satisfaction with our revision and gave positive feedback on our contributions.
Our work is the first to investigate fine-grained shadow detection by considering opacity variations explicitly, which can offer important insights into occluder distance and light source intensity, and benefit a wide range of scene understanding applications.

We are grateful to receive positive recognition as follows:
- **New problem and dataset (i.e., Fine-grained Shadow Detection)** [Reviewers Xc6q, DKcE, cS6y, WyPt]
- **Comprehensive experiments** [Reviewers DKcE, cS6y, WyPt]
- **Clear presentation** [Reviewers Xc6q, cS6y, WyPt]

The main concerns are summarized as follows:
- **More discussions on downstream applications.**
As we make the first attempt towards the new fine-grained shadow detection task, we agree with reviewers that additional analysis would strengthen our work.
We have added more quantitative comparisons and user studies on downstream tasks like shadow removal and 3D reconstruction in the rebuttal.

- **More baselines and ablation studies.**
As suggested, we have included more baselines and ablation studies in the rebuttal to further validate the effectiveness of our solution.

- **Detailed dataset description.**
We have highlighted the key features of our dataset in the rebuttal, and will improve the dataset description.
Our dataset and code will also be released to facilitate future research in the community.

Following the updates in the rebuttal, reviewers acknowledged that our responses have addressed their main concerns, with several noting that they would raise their scores.

We would like to express our sincere gratitude again to the reviewers for their constructive comments and suggestions.
Given the opportunity, we will incorporate all suggestions with details in the revision.

---

### Decision · Program_Chairs · 2025-09-17

**Decision:**

Accept (poster)

**Comment:**

This paper initially received mixed to slightly negative ratings, with most reviewers leaning toward rejection or borderline accept due to concerns about novelty, clarity of motivation, missing comparisons, and limited evaluation. After the rebuttal, however, all reviewers acknowledged that the authors had addressed most of these concerns satisfactorily. Several reviewers explicitly noted that the rebuttal resolved their key issues, though they still encouraged the authors to strengthen the paper with additional results, updated comparisons, and deeper analysis.

Overall, while the paper remains borderline in its current form, the consensus is that the idea is technically sound and the contributions are promising. The authors should revise the paper accordingly for the camera-ready version to include experiments or discussions in the rebuttal and fully address the remaining concerns.